

# Flowing wells: history and role as a root of groundwater hydrology

Xiao-Wei Jiang[1,*], John Cherry[2], Li Wan[1]

1. MOE (Ministry of Education) Key Laboratory of Groundwater Circulation and Evolution, China University of Geosciences, Beijing 100083, China

2. G360 Institute for Groundwater Research, University of Guelph, Guelph, Ontario N1G 2W1, Canada

*Correspondence to: Xiao-Wei Jiang (jxw@cugb.edu.cn)

## Abstract

The spewing of groundwater in flowing wells is a phenomenon of interest to the public, but little attention has been paid to the role of flowing wells on the science of groundwater. This study reviews that answering to problems related to flowing wells since the early 19[th] century led to the birth of many fundamental concepts and principles of groundwater hydrology. The concepts stemmed from flowing wells in confined aquifers include permeability and compressibility, while the principles include Darcy's law, role of aquitards on flowing well conditions and the piston flow pattern, steady-state well hydraulics in confined aquifers, and transient well hydraulics towards constant-head wells in confined or leaky aquifers, all of which are applicable even if flowing well conditions have disappeared. Due to the widespread occurrence of aquitards, there is a long-lasting misconception that flowing wells must be geologically-controlled. The occurrence of flowing wells in topographic lows of unconfined aquifers was anticipated in 1940 and later verified in the 1960s, accompanying with the birth of the theory of topographically-driven groundwater flow, which has been considered as a paradigm shift in groundwater hydrology. Based on studies following this new paradigm, several preconditions of flowing wells given in the 19[th] century have been found to be not necessary at all. This historical perspective of the causes of flowing well conditions and the role of flowing wells on the science of groundwater could lead to a deeper understanding of the evolution of groundwater hydrology.



## 1 Introduction

The primary motivation for the study of groundwater is its role as a resource (Back and Herman, 1997;Freeze and Cherry, 1979). Water from springs was utilized by people in the Middle East about 10,000 years ago (Beaumont, 1973), while from shallow flowing wells, which overflow at the surface, was used in northern France as early as 1126 (Margat et al., 2013). Davis and De Weist (1966) pointed out that exploration of flowing wells in Europe in the 18th century was responsible for stimulating the advancement of water well drilling technology. It was believed that the emergence of groundwater hydrology (hydrogeology) as a distinct science in the 19th century was a result of the maturation of its mother sciences (geology and hydrology) in the 19th century (Fetter, 2004;Meyer et al., 1988). Although it has been realized that flowing wells have always attracted considerable public interest (Chamberlin, 1885;Freeze and Cherry, 1979;Meiter, 2019), little attention has been paid to the role of flowing wells on the science of groundwater.

Due to the function of providing clean groundwater without pumping, flowing wells were a significant source of water supply for drinking and/or agriculture in Europe and the United States in the 19th century and early 20th century (Howden and Mather, 2013;Konikow, 2013). During this period, in the process of exploring flowing wells and observing the flow rates in flowing wells, field observations of flowing wells led to some fundamental concepts and principles of groundwater hydrology in Europe and the United States. For example, based on field conditions in such regions as Paris, London, Wisconsin, and North and South Dakotas, the qualifying conditions of flowing wells were well recognized (Chamberlin, 1885;Darton, 1905;Bond, 1865;Darton, 1897), which further led to the pattern of piston flow in confined aquifers; inspired by observations of flow rates of flowing wells in Paris, Darcy (1856) identified the controlling factors of flow in subsurface media and Dupuit (1863) established the principles of steady-state groundwater flow (Ritzi and Bobeck, 2008); based on the decreasing discharge in flowing wells with time and the excess of groundwater discharge compared to estimates of groundwater recharge, Meinzer (1928) identified the compressibility of confined aquifers, which constitutes the basis of transient groundwater flow. Probably because these concepts and principles are applicable to non-flowing wells and most developments on well hydraulics since the 1930s were based on pumping in non-flowing wells, it seems that the role of flowing wells on the development of these concepts and principles in history was not realized in current textbooks.





The widespread occurrence of confined aquifers and the accompanying phenomenon of flowing wells in the initial stage of groundwater development made the piston flow pattern and geologically-controlled flowing wells (Fig. 1a) a widely accepted conceptual model in groundwater hydrology. By analyzing the cross-sectional flow pattern from recharge to discharge areas in thick unconfined aquifers,

Hubbert (1940) pointed out that a confined aquifer outcrops in the highlands and overlain by impermeable strata in the lowlands as shown in Fig. 1a is by no means a necessary condition for flowing wells, and found that flowing wells could occur in topographic lows without an overlying confining bed (Fig. 1b). In the course of exploring groundwater in the Canadian Prairies, Tóth (Tóth, 1962, 1963, 1966) and Meyboom (1962; 1966) verified the occurrence of topographically-controlled flowing wells in the

field and further developed Hubbert's model of topographically-driven groundwater flow from recharge to discharge areas. In the Canadian Prairies, one of the topographically-controlled flowing wells has a well depth of only 9 m (Tóth, 1966). Kasenow (2010) reported a topographically-controlled flowing well in the discharge area of an unconfined aquifer, which was constructed at a depth of only 6.1 m below surface but has a head of around 1.4 m above surface. The principles of topographically-driven

groundwater flow systems and the cause of topographically-controlled flowing wells constitute a paradigm shift in groundwater hydrology (Bredehoeft, 2018;Madl-Szonyi, 2008). Unfortunately, topographically-controlled flowing wells, which is a natural consequence of topographically-driven groundwater flow, are described only in very few textbooks (Domenico and Schwartz, 1998;Freeze and Cherry, 1979;Heath, 1983;Kasenow, 2010), the number of which is very limited compared with the large

number of groundwater textbooks. In other words, although the concept of flowing well is introduced in almost every groundwater textbook, a complete understanding of the causes of flowing wells is still missing in most textbooks. This also undermines the role of topographically-controlled flowing wells on the paradigm shift from the conceptual model of piston flow to topographically-driven flow.

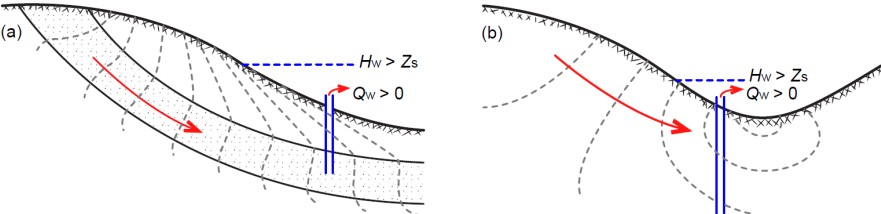

Fig. 1 Geologically-controlled (a) and topographically-controlled (b) flowing wells (Modified from Freeze and Cherry, 1979).



Freeze and Back (1983) divided physical hydrogeology into three domains, i.e., physics of groundwater flow, well and aquifer hydraulics, and regional groundwater flow. Following this approach, studies directly related to flowing wells or bridge classical studies on flowing wells are divided into four

threads (Fig. 2), which cover all of the three domains. The three threads shown in Fig. 2a-c are all stemmed from geologically-controlled flowing wells, while the thread shown in Fig. 2d is from topographically-controlled flowing wells. Note that the seven classical studies which were selected by both Freeze and Back (1983) and Anderson (2008) as benchmark papers of groundwater hydrology (physical hydrogeology) have been included in the four threads, and four out of the seven papers are

directly related to flowing wells, implying that flowing wells can be considered as, at least, one of the roots of groundwater hydrology. The main aim of this review is to give a clear picture on the history of drilling flowing wells and the role of flowing wells on the evolution of the four threads of physical hydrogeology.

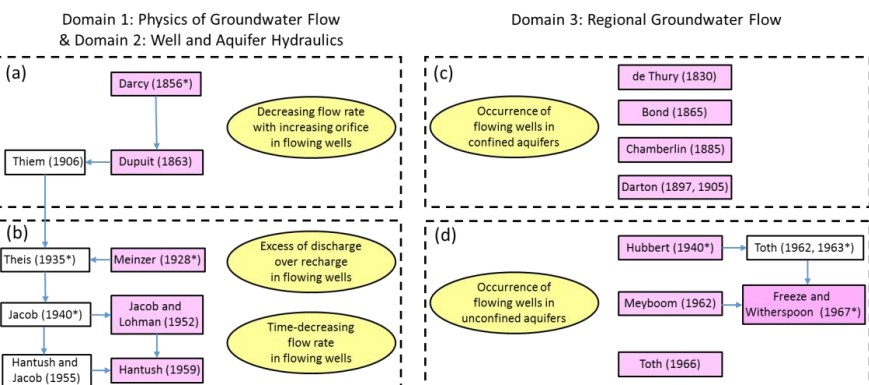

Fig. 2 Four threads of evolution of groundwater hydrology stemmed from flowing wells. The yellow ellipses are field phenomena of flowing wells, the purple boxes are papers directly related to flowing wells, and the white boxes are papers less or not directly related to flowing wells but have connections to previous or follow-up studies on flowing wells. The publications labeled with a "*" are included in both Freeze and Back (1983) and Anderson (2008).


The paper is organized as follows. We first introduce the terms used to represent flowing wells, and the evolution of the ambiguous term "artesian well", which was initially used to represent flowing wells in confined aquifers but was later widely used to denote flowing well in both confined and unconfined aquifer, or any well penetrating a confined aquifer (Sect. 2). After introducing the history of drilling



flowing wells in selected regions that had inspired groundwater hydrologists in Sect. 3, the four threads

of evolution of groundwater hydrology rooted from flowing wells are briefly summarized in Sect. 4

through 7, which are sequenced based on the order of earliest publications about each thread. Finally,

some concluding remarks are given in Sect. 8.

## 2 Terms related to flowing wells and definitions of "artesian"

**2.1 Terms representing flowing wells**

Photos of flowing wells were used as the cover image or frontispiece of some professional

publications (Chamberlin, 1885;Freeze and Back, 1983;Hudak, 2005;Deming, 2002;Younger, 2007),

which indicates the interest of groundwater professionals in flowing wells. In groundwater hydrology,

hydrogeology or hydrology textbooks available to the authors, we found the phenomenon of a well which

overflows at the surface is defined or mentioned in 34 textbooks. The widely used terms include "flowing

well", "artesian well" and "flowing artesian well", which appear in 17, 15 and 13 textbooks, respectively

(sometimes more than one term is used in a book). Other less frequently used terms include "artesian

flowing well", "overflowing well", and "free flowing well". For convenience of discussion, we divide

the six terms into two categories based on whether "artesian" is shown.

The terms "flowing well", "overflowing well" and "free flowing well" stem purely from the

phenomenon of water overflow at the well outlet. The term "overflowing well" has been used in Britain

since at least 1820s (Anonymous, 1822), and currently is still widely used in Britain, as found in several

textbooks (Hiscock and Bense, 2014;Price, 1996;Rushton, 2003). To the authors' knowledge, the term

"flowing well" was first used in the USGS hydrogeologic report *Requsite and Qualifying Conditions of*

*Artesian Wells* (Chamberlin, 1885) and is currently the most widely used one. The term "free flowing

well" can be found in three textbooks (Fitts, 2013;Kruseman and de Ridder, 1990;Nonner, 2003). By

including the adjectives "flowing", "overflowing" or "free flowing", these terms have clear meaning to

represent wells that do not need pumping.

The number of textbooks that use one or two of the terms "artesian well", "flowing artesian well"

and "artesian flowing well" is as high as 28, indicating the popularity of the adjective "artesian".

"Artesian well" derived its name from the place Artois ("Artesia" is the historical Latin name of Artois)

where the first flowing wells were obtained in the 12[th] century, and gives birth to the terms "flowing

artesian well" and "artesian flowing well" by adding the adjective "flowing". Currently, in the majority





of textbooks in Europe (Hendriks, 2010;Kruseman and de Ridder, 1990;Price, 1996;Rushton, 2003;Brassington, 2017;Davie, 2008;de Marsily, 1986;Hölting and Coldewey, 2019) and in at least eight textbooks in North America (Deming, 2002;Domenico and Schwartz, 1998;Driscoll, 1986;Pinder and Celia, 2006;Yeh et al., 2015;Hornberger et al., 2014;Fitts, 2013;Alley and Alley, 2017) , the term "artesian well" is synonymous with flowing well. Note that in ten textbooks in North America (Fetter, 2001;Freeze and Cherry, 1979;Batu, 1998;Kasenow, 2010;LaMoreaux et al., 2009;Mays, 2012;McWhorter and Sunada, 1977;Heath, 1983;Schwartz and Zhang, 2003;Todd and Mays, 2004), an artesian well means a well that derives its water from a confined aquifer, the details of which are discussed in Subsect. 2.2 and 2.3.

Before 1940s, it was believed that only a confined aquifer has the possibility to have hydraulic head higher than the ground surface elevation, i.e., flowing wells could occur only in confined aquifers. Hubbert (1940) first noted that flowing wells could occur in the discharge area of a homogeneous basin (Fig. 1b, the details can be found in Subsect.7.1). This explanation of the cause of flowing wells was accepted by the USGS (Heath, 1983;Lohman, 1972a;Lohman, 1972b). In Heath (1983) and Lohman (1972b), the term "flowing artesian well" was restricted to flowing wells in confined aquifers, and in Heath (1983), it was explicitly pointed out that "a flowing well does not necessarily indicate artesian conditions". To differentiate the two types of flowing wells due to different causes, Freeze and Cherry (1979) defined geologically-controlled flowing wells and topographically-controlled flowing wells. As shown in Fig. 1, the former develop in confined aquifers and receive recharge at upland outcrops, while the latter occur in the topographic lows of unconfined aquifers.

**2.2 Evolution of "artesian well" and the birth of "flowing artesian well"**

Literally, "artesian well" stands for "well of Artois". It is unquestionable that it was the phenomenon of water overflow at the surface which attracted people's attention to wells of Artois (Fuller, 1906;Norton, 1897). As early as 1805, the name "artesian fountain" was applied in French scientific literature to represent flowing wells (Lionnais, 1805). In later publications, "artesian well" was widely used to represent flowing wells in France and Britain (Arago, 1835;de Thury, 1830;Garnier, 1822;Buckland, 1836). The term "artesian well" was introduced in the United States in 1835 (Storrow, 1835). In Chamberlin (1885), the terms "artesian well", "artesian fountain" and "flowing well" were used interchangeably.





Probably because most artesian wells were much deeper than traditional dug wells, and a driller could not assure a deep well could overflow at the surface before finishing drilling, the term "artesian well" was frequently used to denote a deep well that did not overflow. Chamberlin (1885) condemned such a use, however, in less than 20 years, Chamberlin and Salisbury (1904) pointed out that "*at present time any notably deep well is called artesian, especially if it descends to considerable depths*". Moreover, because deep wells or artesian wells were drilled instead of being dug, artesian wells were also widely used to denote drilled/bored well in the 19th century. Fortunately, such usages were seldom adopted in subsequent years, probably due to the contribution of definitions given in Meinzer (1923b).

As the geologic conditions governing flowing wells became known, the term "artesian well" was suggested to represent any well in which hydraulic head is higher than the elevation of water table in the 1890s (Carpenter, 1891;Norton, 1897;Slichter, 1899). Norton (1897) insisted that if there are two wells in the same town derived from the same aquifer and rising to the same height, one could overflow at the surface but the other could not just because of slightly higher ground surface, it was preferable to term both wells artesian wells. Such a usage was accepted and popularized by the USGS (Lohman, 1972b;Meinzer, 1923b). In some later publications by authors of the USGS, the term "artesian well" was used equivalently to a well in a confined aquifer (Jacob, 1940, 1946, 1947;Meinzer, 1928;Theis, 1935).

Since the 19th century, artesian wells were not restricted to a flowing wells and can be divided into flowing artesian wells and non-flowing artesian wells. Although the term "non-flowing artesian well" (or "negative artesian well") was used in the 19th century and early 20th century (Arago, 1835;Norton, 1897;Slichter, 1899;Meinzer, 1928), the adjective "non-flowing" is used only in very limited textbooks (Heath, 1983;Schwartz and Zhang, 2003;Singhal and Gupta, 2010). In the textbooks that define an artesian well to be any well tapping a confined aquifer, an artesian well is by default a non-flowing artesian well (Fetter, 2001;Todd and Mays, 2004;Abdrashitova, 2015;Kasenow, 2010).

**2.3 Confusing uses of "artesian" in the literature**

The adjective "artesian", which can be traced to the terms "artesian fountain" and "artesian well", has been applied to such terms as "artesian pressure", "artesian water" and "artesian aquifer". Following the definition given by the USGS (Lohman, 1972b), artesian water and artesian aquifer are equivalent to confined groundwater and confined aquifer, artesian pressure can be considered as the pressure in a confined aquifer, and an artesian well is a well that derives water from a confined aquifer. Such



definitions of artesian well and artesian aquifer can be found in 14 textbooks (Abdrashitova, 2015;Batu, 1998;Fetter, 2001;Heath, 1983;Lohman, 1972a;Mays, 2012;McWhorter and Sunada, 1977;Rudakov, 2014;Schwartz and Zhang, 2003;Sen, 2015;Singhal and Gupta, 2010;Todd and Mays, 2004;Kasenow, 2010;LaMoreaux et al., 2009). Note that the majority of these authors are from the United States and Canada, with four exceptions: Russia (Abdrashitova, 2015) , Ukraine (Rudakov, 2014), India (Singhal and Gupta, 2010) and Turkey (Sen, 2015).

However, in other 20 textbooks (Alley and Alley, 2017;Bear, 1972, 1979;Brassington, 2017;Davie, 2008;de Marsily, 1986;Deming, 2002;Domenico and Schwartz, 1998;Driscoll, 1986;Hendriks, 2010;Hölting and Coldewey, 2019;Hornberger et al., 2014;Kruseman and de Ridder, 1990;Nonner, 2003;Price, 1996;Fitts, 2013;Pinder and Celia, 2006;Yeh et al., 2015;Rushton, 2003;Dassargues, 2019) that defined one or several of the terms "artesian pressure", "artesian water", "artesian well" and "artesian aquifer", the water level in the artesian well should be higher than the ground surface, and at least part of an artesian aquifers should have hydraulic head above the ground surface. Note that the authors of eight of these books are from the United States.

Many basins throughout the world were called artesian basins. The Great Artesian Basin in Australia is one of the largest artesian basins in the world and is well known for its numerous flowing wells in confined aquifers. In the United States, many basins were termed artesian basins, for example, the artesian basin of the Dakotas (Darton, 1905;Swenson, 1968), the great Paleozoic artesian basin of the Mississippi Valley region (Meinzer, 1923a), the Rowsell artesian basin in New Mexico (Fiedler and Nye, 1933), and the Grand Junction artesian basin (Jacob and Lohman, 1952). According to Meinzer (1923b), an artesian basin is a geologic structural feature or combination of such features in which water is confined under artesian pressure, implying that the hydraulic head being greater than the elevation of ground surface is not a necessary condition of an artesian basin . In fact, all of these well-known artesian basins have many flowing wells in the initial stage of groundwater development. Therefore, it is difficult to interpret the meaning of the adjective "artesian" in the term "artesian basin".

The different meanings of "artesian" caused confusion not only to beginners of groundwater hydrology, but also to professional groundwater hydrologists. The confusion caused by "artesian " has been realized by some textbook authors. Deming (2002) and Younger (2007) both chose photos of a flowing well for their cover image, but they have quite opposite viewpoints on "artesian". Deming (2002) from the United States held the opinion that "artesian" implies that the hydraulic head is greater than the


elevation of ground surface, and defining "artesian aquifer" to be identical to confined aquifer would make the definition not only conceptually useless, but also etymologically incorrect because wells drilled in Artois in 1126 could flow spontaneously. On the contrary, Younger (2007) from the United Kindom

believed that "artesian" is a synonym of "confined", and pointed out that "artesian" is also widely misused to refer to any well from which water flows without pumping, a phenomenon which is not restricted to confined aquifers. Younger also discouraged further use of "artesian" because it lacks intuitive meaning in modern English.

To sum up, hydrogeologists are keenly interested in flowing wells, but are also confused by the

term "artesian". This confusion leads to underestimation of the role of flowing wells on the development of groundwater hydrology. In the following discussion, we avoid using the confusing adjective term "artesian well". Instead, following Freeze and Cherry's (1979) classification of flowing wells, we use geologically-controlled flowing wells to represent flowing wells in confined aquifers or leaky aquifers, and topographically-controlled flowing wells for wells in aquifers without a confining bed.

## 235 3 The history of drilling flowing well in selected regions

Many aquifers had hydraulic heads above the land surface when the first deep wells were drilled (Fetter, 2001). A thorough review on the history of flowing well drilling is beyond the scope the current discussion. Here, we briefly review the history of flowing well in regions that directly inspired hydrogeologists.

### 240 3.1 France

As early as 1126, the first shallow flowing well tapping the confined fringe of the chalk aquifer was obtained in Artois in northern France (Margat et al., 2013). The technique of cable-tool drilling (also called percussion drilling) resulted in drilling of deeper flowing wells in France in the early 19th century. Garnier (1822) published the first technical guidebook on drilling artesian wells. It was stated that with

the exception of some provinces, there are few parts of France where artesian wells might not be procured. Garnier obtained a prize from the Society for the Encouragement of Industry due to the publication of this book, which reflects the interest of the French government in such wells.

According to Arago (1835), most flowing wells up to that time ranged in depth from 36 m to 177 m, however, one borehole which was drilled in quest of coal resulted in the formation of a flowing well

with a depth of 314 m. Between 1833 and 1841, a flowing well with a depth of 548 m was drilled in

Grenelle within the Paris Basin. This flowing well could rise to a height of 33 m above ground surface in a pipe supported by a wooden scaffolding which was accessible by steps (Fig. 3). In 1855, an article named *The Artesian Well at Grenelle, in France* was reproduced in *California Farmer and Journal of Useful Sciences* in the United States (Anonymous, 1855). It was commented that "*This splendid*

*achievement at that date may be looked upon as the pioneer effort, and at the present time and within a very few year, the most astonishing results may be expected.*" In 1861, another flowing well with a depth of 586 m was finished in Passy near Paris, which is the last flowing well in Paris that is still in use today. Darcy (1856) and Dupuit (1863) were both inspired by flow rate measurements at different orifices in these deep flowing wells. The details are given in Sec. 5.

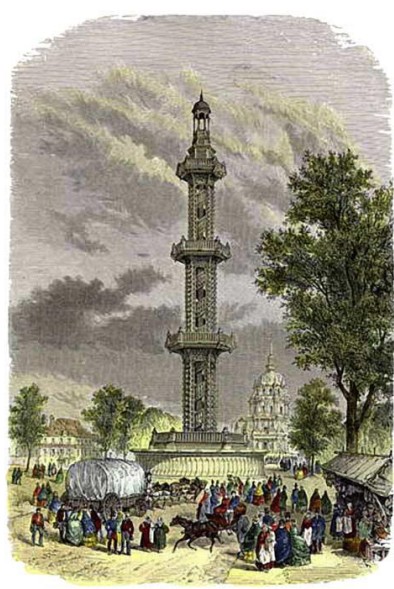


Fig. 3 The wood engraving of the flowing well at Grenelle (uncopyrighted and freely distributed by http://www.antiqueprints.com). The external structure was demolished in 1903 (Ritzi and Bobeck, 2008).


### 3.2 Great Britain

In Great Britain, James Ryan obtained a patent on boring for minerals and water in 1805, while John Goode obtained a patent on boring for the purpose of obtaining and raising water in 1823 (Macintosh, 1827). The grant of these patents indicates that Great Britain was active in drilling wells in the early 19[th]

century. In 1807, it was reported that there were flowing wells in the Thames Basin, London (Farey,





1807). According to an article in *Monthly Magazine and British Register* (Anonymous, 1822), many flowing wells had existed for a long time in and near London, and in various parts of the country by 1822, and two flowing wells drilled in a town called Tottenham in 1821 had depths of 32 m and 37 m.

The publication of Garnier (1822) in France aroused further interest in Great Britain (Farey, 1823). By 1840, many "artesian wells" had been drilled in the London Basin (Mylne, 1840), although many boreholes failed because of lack of geological information. The experience gained from the costly failures improved understanding of conditions necessary for the success of a flowing well, the details of which are to be discussed in Subsect. 4.1.

### 3.3 The United States

To meet the water supply in some cities as well as irrigation demands in farms, numerous flowing wells were drilled in the United States beginning in the 19th century as a direct result of the increased drilling technology. Here, we list some regions where flowing wells were drilled, causing significant advances in groundwater hydrology.

Development of the Cambrian-Ordovician aquifer system in the northern Midwest can be traced to 1864 when a flowing well with a depth of 217 m was drilled in Chicago (Konikow, 2013). By the end of the 19th century, flowing wells were common in topographically low areas in the Mississippi, Missouri, and Illinois River valleys, near Lake Michigan, and around Lake Winnebago in northeastern Wisconsin (Young, 1992). At the beginning of the 21st century, there were still many flowing wells newly drilled in Michigan (Gaber, 2005). Chamberlin's (1885) classic report was based on the hydrogeologic conditions in Wisconsin, and is considered as one of the roots of groundwater hydrology in Wisconsin (Anderson, 2005). In 1876, a flowing well with a depth of 293 m and an initial flow rate of 3270 m$^3$/d, which was named *The Greatest Artesian Fountain in America*, was drilled in Prairie du Chien, Wisconsin (Meiter, 2019). The photo of this flowing well was used as the frontispiece of Chamberlin (1885) and Freeze and Back (1983), the cover image of Deming (2002), and was also cited in Anderson (2005).

In the Great Plains, interest in groundwater emerged due to the irrigation demands beginning in the 1880s, due to the widespread drought. In early 20th century, flowing wells were common in topographic lows near rivers, for example, in the Arkansas River valley of southeastern Colorado, much of South Dakota, and parts of southeastern North Dakota and northeastern Nebraska in the Missouri River valley (Darton, 1905). In South and North Dakota within the Great Plains, there were about 400 deep wells

drilled to the Dakota sandstone by 1896, of which over 350 were flowing wells (Darton, 1897). Due to

the introduction of the jetting method of drilling in around 1900, thousands of small-diameter wells were

drilled to the Dakota sandstone during the following two decades. There were about 10,000 deep wells

in South Dakota in 1915, and between 6,000 and 8,000 deep wells in North Dakota in 1923 (Meinzer and

Hard, 1925). Due to the increased withdrawal of deep groundwater, many flowing wells became non-

flowing wells, accompanied by decreasing flow rates in the flowing wells that still flowed. The condition

of flowing wells led to improved understanding of the pattern of groundwater circulation in confined

aquifers (Darton, 1905), while the imbalance between groundwater discharge and recharge led to the

birth of the concept of compressibility of confined aquifers (Meinzer and Hard, 1925; Meinzer, 1928).

In flowing wells of the Dakota aquifer, it was noted that "*the pressure increases for several hours

*or even days after the flow is shut off, and when opened the flow decreases in the same way until the*

*normal flow is reached*" (Meinzer, 1928). Several decades later, based on field observations of decreasing

flow rate with time in flowing wells in the Grand Junction artesian basin and in the Rowsell artesian

basin in New Mexico, constant-drawdown aquifer tests were proposed to obtain hydraulic parameters

(Jacob and Lohman, 1952;Hantush, 1959).

**3.4 Australia**

The Great Artesian Basin, which covers 1/5 of the total area of Australia, is one of the largest and

well-known groundwater basins in the world (Ordens et al., 2020). The first shallow flowing well with a

depth of 43 m was dug by using an auger near a spring in New South Wales in 1878, while the first

drilled flowing well with a depth of 393 m was completed near Cunnamulla, Queensland in 1887

(Williamson, 2013). By the end of the 19[th] century, there were already around 1000 flowing wells (van

der Gun, 2019). The discovery of flowing wells and "artesian water" triggered the emergence of

hydrogeology as a discipline in Australia (Williamson, 2013), and the development of "artesian water"

have played a vital role on the pastoral industry in the arid and semi-arid regions of Australia (Habermehl,

2020).

Due to the occurrence of intervening aquifers and aquitards, the Great Artesian Basin is a multi-

layered confined aquifer system. Although head drawdowns of up to 100 m have been recorded in highly

developed areas, hydraulic heads in the Jurassic and Lower Cretaceous aquifers are still above ground

surface throughout most of the basin (Habermehl, 2020). In Australia, currently the term "artesian" still





implies that a bore will flow naturally (Williamson, 2013). A comprehensive review of the history and

recent research status of the basin can be found in Ordens et al. (2020).

**3.5 Canada**

Since the beginning of the 20th century, the hydrogeology of the Canadian Prairies has been studied.

Groundwater in this region is obtained from surficial Pleistocene glacial drift and from the underlying

bedrock of Tertiary or Cretaceous age. Quaternary glacial deposits and the underlying Tertiary Paskapoo

sandstone constitute a thick unconfined aquifer. A similarity between the potentiometric surface and the

local topography were widely observed in many parts of the Canadian Prairies (Jones, 1962;Meyboom,

1962;Tóth, 1962;Farvolden, 1961). Due to the occurrence of a large number of flowing wells, either in

the glacial drift or in the bedrock, great attention was paid to the relation between topography, geology

and areas with flowing wells during basin-scale groundwater surveys (Meyboom, 1966).

The Trochu area in central Alberta, which covers an area of 67 km$^2$, is representative of the

hydrogeology of Canadian Prairies. There were 10 shallow flowing wells ranging in depth from 9 m to

27 m in topographic lows (Tóth, 1966). Because the glacial deposits have low contents of clay, they are

efficient for infiltration of rainfall and evaporation of soil water. Therefore, the Quaternary glacial

deposits and the underlying Tertiary Paskapoo sandstone constitute a thick unconfined aquifer.

Combined with previous theoretical findings on topographically-driven flow systems (Hubbert,

1940;Tóth, 1962, 1963), these flowing wells in unconfined aquifers were considered to be controlled by

topography and are typical manifestations of groundwater discharge (Tóth, 1966). The details are

discussed in Sect. 7.

## 4 Geologically-controlled flowing wells and piston-flow in confined aquifers

**(1820s-current)**

**4.1 Conditions of geologically-controlled flowing wells**

Due to the progress of hydrology and geology in the 18$^{th}$ century, it was accepted in the early 19$^{th}$

century that the water of flowing wells came from rainfall, which found its way through the pores or

fractures of a permeable stratum enclosed between two water-tight strata (Garnier, 1822). de Thury (1830)

summarized three conditions of flowing wells in confined aquifers. The first is to reach a flow of deep

water coming from higher basins and passing along the bosom of the earth between compact and



impermeable rocks; the second is to afford this deep water the possibility of rising to the surface by using an artificially bored well; and the third is to prevent the spreading of the ascending water into the surrounding sand or rock by inserting tubes into the bored well.

Following the successful drilling of flowing wells in France, the theory behind the occurrence of flowing wells was well understood in Britain. Buckland (1836) illustrated the cause of two flowing wells in the confined aquifer of the London Basin. The successful experiences coupled with the costly failures of drilling in Britain led to the recognition of three necessary conditions for the success of a flowing well (Bond, 1865). The conditions are:

*"(1) The existence of a porous stratum having a sufficient outcrop on the surface to collect an adequate amount of rainfall, and passing down between two impermeable strata; (2) the level of the outcropping portion of the porous stratum must be above that of the orifice of the well, so as to give a sufficient rise to the water; (3) there must be no outlet in the porous stratum by which its drainage can leak out, either in the shape of a dislocation, by which it can pass into lower strata,*

*or a natural vent, by which it can rise to the surface at a lower level than the well."*

Based on his experience in the Wisconsin part of the Cambrian-Ordovician aquifer system, Chamberlain (1885) published *The Requisite and Qualifying Conditions of Artesian Wells* and listed seven conditions of flowing wells. The conditions include:

"*(1) A pervious stratum to permit the entrance and the passage of the water; (2) A water-tight bed*
*below to prevent the escape of the water downward; (3) A like impervious bed above to prevent escape upward, for the water, being under pressure from the fountain-head, would otherwise find relief in that direction; (4) An inclination of these beds, so that the edge at which the waters enter will be higher than the surface at the well; (5) A suitable exposure of the edge of the porous stratum, so that it may take in a sufficient supply of water; (6) An adequate rain-fall to furnish this supply;*
*(7) An absence of any escape for the water at a lower level than the surface at the well.*"

In fact, the seven prerequisites given by Chamberlain (1885) are more or less similar to the three conditions given by de Thury (1830) and Bond (1865) several decades earlier. However, Bonds (1865) did not cite de Thury (1830), and Chamberlin (1885) did not cite either Bonds (1865) or de Thury (1830), therefore, we assume their findings were obtained independently. Chamberlin (1885) also pointed out
that confining layers are not totally impermeable, which foreshadowed later studies on well hydraulics of leaky aquifers (Hantush, 1959;Hantush and Jacob, 1955;Jacob, 1946) and leakage in sedimentary



basins (Swenson, 1968). Chamberlin's report was recognized as the first classic paper on regional groundwater flow in the United States (Bredehoeft et al., 1982).

**4.2 Piston flow in confined aquifers**

390       In the late 1890s, Darton (1897) investigated the occurrence of flowing wells in the Dakotas and plotted the cross-section of the Dakota aquifer (Fig. 4). By constructing the hydraulic head contours of the Dakota aquifer in South Dakota, which shows the head loss through the confined aquifer, Darton (1905) concluded that groundwater discharged by the flowing wells in the east had flowed hundreds of kilometers through the confined aquifer from the outcrops in the west. This study popularized the pattern

of groundwater flow in a confined aquifer which outcrops in topographic highs as shown in Fig. 1a. In the Great Artesian Basin of Australia, there was also a long-lasting conceptual model that each aquifer can be considered to be laterally continuous across the extent of the basin (Habermehl, 2020). In the Dakota confined aquifer, vertical leakage into the aquifer from adjacent strata was identified in the 1960s (Swenson, 1968). In Australia, several recent studies also identified vertical connections between

aquifers (Pandey et al., 2020;Smerdon and Turnadge, 2015). However, vertical leakage seldom change the direction of groundwater flow within the confined aquifer.

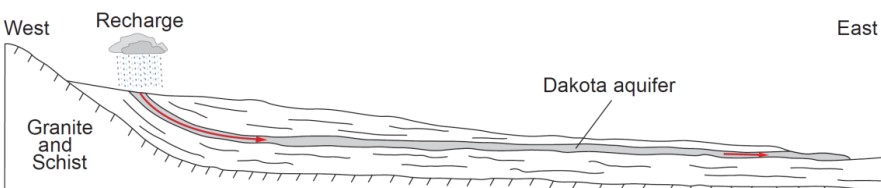

Fig. 4 The profile of the Dakota aquifer and the piston flow in a confined aquifer. (Modified from Darton,
405   1897)

      As shown in Fig. 4, the flow pattern in a confined aquifer is similar to flow through a pipe, and is commonly referred to as piston flow (Bethke and Johnson, 2008;Hinkle et al., 2010). Because many

hydrogeochemical processes are dependent on travel time through the aquifer, hydrochemical facies (Back, 1960; 1966) usually evolve along the flow path within the piston flow. Therefore, the piston flow model, which stemmed from analyzing geologically-controlled flowing wells, is the cornerstone of sampling and analyzing groundwater geochemistry and isotopes in confined or leaky aquifers.



## 5 Darcy's law and steady-state well hydraulics inspired by flowing wells (1850s-

**1910s)**

### 5.1 The birth of Darcy's law evoked by flowing wells

It is widely known that Darcy's law, which represents the beginning of groundwater hydrology as a quantitative science (Freeze and Back, 1983), was established based on sand column experiments. In fact, the sand column experiments were designed to confirm a linear correlation between flow rate and

head loss in sands which was discovered in flowing wells (Brown, 2002;Ritzi and Bobeck, 2008). Because flowing wells were important sources of water supply in the Paris Basin, flow rates at different elevations of discharge orifices were measured in several flowing wells in the 1840s (Fig. 5), which were called experiments of head loss versus riser pipe height (Ritzi and Bobeck, 2008). In fact, such experiments can be regarded as constant-head (drawdown) tests in single wells. As shown in Fig. 5a, the

flow rates measured in September and November increased linearly as the elevation of discharge orifice decreased. The higher flow rate in November can be a result of higher hydraulic head surrounding the flowing well, probably due to the contribution of groundwater recharge. Darcy was interested by this linear correlation (Brown, 2002;Ritzi and Bobeck, 2008).

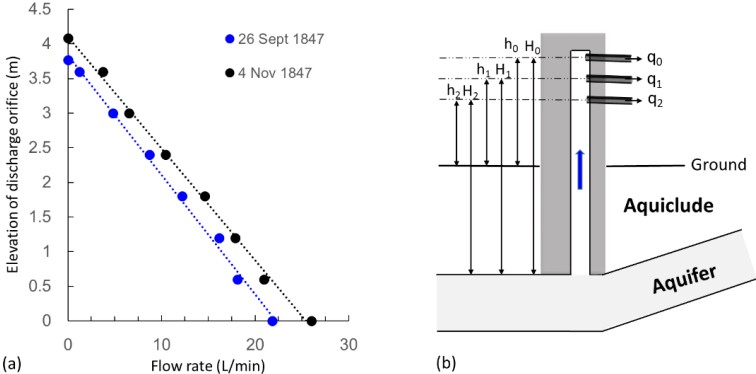

Fig. 5 The change in flow rate with elevation of discharge orifice of a flowing well (a) and the measuring device (b). (Data shown in a is from Darcy, 1856)

Theoretically, the head loss from the recharge area to the discharge orifice can be divided into head loss in the aquifer and head loss along the well pipe. According to the Chezy equation describing head

loss at high flow velocities, head loss in the well pipe should be proportional to the square of flow rate.





Based on the linear trend between head loss and flow rate shown in Fig. 5a, it was inferred that the head

loss in the short-distance well pipe with high velocities is limited compared with the well loss in the long-

distance aquifer with low velocities. To identify the control of flow velocity on head loss, Darcy

conducted pipe flow experiments during 1849 and 1851 and proposed equations on the dependence of

head loss on flow rate. At low velocity, the head loss was found to be linear to discharge rate, which can

be written as

$$h_L = \frac{La}{\pi r^4} q \tag{1}$$

and at high water velocity, the head loss was found to be linear to the square of discharge rate, which can

be written as

$$h_L = \frac{Lb}{\pi r^5} q^2 \tag{2}$$

where $L$ is length, $r$ is pipe radius, $q$ is the volume discharge rate, and $a$ and $b$ are empirical coefficients

of proportionality. Eq. (1) verified Poiseuille (1841) equation by experiments under completely different

circumstances, and Eq. (2) can be considered as another form of the Chezy equation. Note that Eq. (2)

led to the co-naming of the Darcy-Weisbach pipe friction formula. About 30 years later, Reynolds (1883)

fully quantified the occurrence and differences between laminar and turbulent flow by introducing the

Reynolds number.

By assuming that head loss from the recharge area to the discharge orifice occurs in both the

aquifer and the well pipe, equations similar to the following forms were obtained:

$$h_1 + \frac{aL'}{\pi r'^4} q_1 + \frac{bH_1}{\pi r^5} q_1^2 \cong h_2 + \frac{aL'}{\pi r'^4} q_2 + \frac{bH_2}{\pi r^5} q_2^2 \tag{3a}$$

$$\left(h_1 - h_2\right) + \frac{b}{\pi r^5}\left(H_1 q_1^2 - H_2 q_2^2\right) \cong -C\left(q_1 - q_2\right) \tag{3b}$$

where $L'$ is flow distance in the aquifer from the recharge area, $r'$ is radius representative of pores in the

aquifer, $H_1$ and $H_2$ are the lengths of well pipes from the bottom to the discharge orifice, and $C$ is an

unnamed constant. The linear relationship between $h_1$-$h_2$ and $q_1$-$q_2$ shown in Fig. 5a indicates that the

second term on the left side of Eq. (3b) is negligible, and $C$ can be interpreted as the slope shown in Fig.

5a. To further confirm that head loss in aquifers is linear to the velocity, in 1855, Darcy conducted the

sand column experiments assuming that water flow through sands is similar to water flow through the

aquifer (Darcy, 1856).

The equation he wrote for fluid motion through sands, with the head loss between two points

separated by a distance *l*, is

$$q = ks \frac{h_L}{l} \tag{4}$$

where *s* is the total cross-sectional area perpendicular to flow, and *k* is a coefficient unnamed by Darcy, today call *k* the hydraulic conductivity. Hubbert (1940) rigorously interpreted hydraulic conductivity and examined Darcy's law in the light of the microscopic Navier-Stokes flow theory, which raised the understanding of Darcy's law to a higher level of sophistication.

**5.2 Steady-state well hydraulics in confined aquifers: Dupuit equation**

In 1850, Jules Dupuit succeeded Darcy as Chief Director for Water and Pavements and started his research on groundwater hydraulics. The field data shown in Fig. 5a triggered Dupuit to quantify the constant *C* in Eq. 3b. Dupuit realized that groundwater flow would radially converge to the flowing well and head loss in the confined aquifer away from the flowing well would form a cone of depression (Fig.

6).

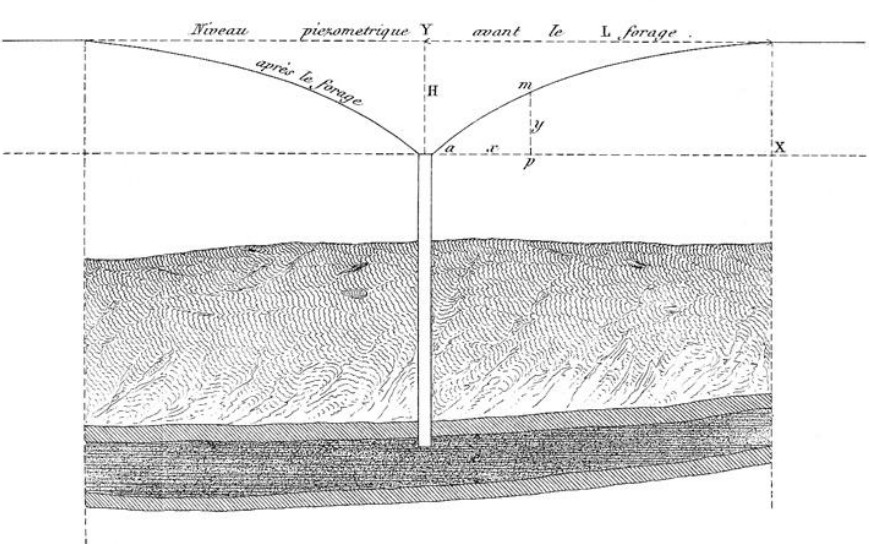

Fig. 6 Plot from Dupuit (1863) showing the radial flow toward a flowing well penetrating a confined aquifer, which caused head loss with a cone of depression in the potentiometric level ("niveau piezometrique"). It is clear that the potentiometric level of the cone of depression is still above the land
surface.

In radial flow, Eq. (4) can be rewritten as





$$q = k\left(2\pi x B\right)\frac{dy}{dx} \qquad (5)$$

where $B$ is the thickness of the confined aquifer, $x$ is the radius within the cone of depression, and $y$ is the corresponding head with reference to the elevation of the discharge orifice. Dupuit (1863) obtained the following equation by integrating equation (5) from the radius of the well, $r_w$, to the radius of influence, $R$:

$$h_0 - h_w = \frac{q}{2\pi k B}\ln\left(\frac{R}{r_w}\right) \qquad (6)$$

where $h_w$ is the elevation of the discharge orifice, and $h_0$ is the hydraulic head at the radius of influence, which equals the initial hydraulic head of the flowing well when the well has been closed for a duration of time. By comparing Eq. (4) and (6), it can be interpreted that $C = \frac{1}{2\pi k B}\ln\left(\frac{R}{r_w}\right)$. Dupuit (1863) explicitly stated that Eq. (6) is supported by the measurements of flow rate versus elevation of discharge orifice of flowing wells reported in Darcy (1856).

Although Eq. (6) was derived based on the hydrogeologic condition of a flowing well in a confined aquifer, it is applicable to non-flowing wells in a confined aquifer. A limitation of the equation is the difficulty of determining the radius of influence in the field. Thiem (1906) improved Eq. (6) by integrating Eq. (5) between two wells within the cone of depression and obtained an equation to determine hydraulic conductivity, which is known as the Thiem equilibrium method. Although the Thiem equation (Thiem, 1906) was introduced in almost all textbooks, Dupuit's pioneering study on steady-state radial flow to a flowing well in confined aquifers (Dupuit, 1863) was seldom mentioned in textbooks.

Dupuit (1863) also derived similar equations for a well in unconfined aquifers by neglecting the vertical hydraulic gradient, which is currently known as Dupuit-Forchheimer approximation. Although the vertical hydraulic gradient is neglected, the Dupuit-Forchheimer approximation is still useful in interpreting regional scale groundwater flow problems (Haitjema and Mitchell-Bruker, 2005).

# 6 Compressibility of confined aquifers and transient well hydraulics inspired by flowing wells (1920s-current)

## 6.1 Compressibility of confined aquifers

In the 1920s, groundwater exploration in the United States had culminated and groundwater





resources were already undergoing development, therefore, much of the effort of the USGS turned toward

inventory (Domenico and Schwartz, 1998). The inventory of groundwater resources in the Dakota aquifer,

where the number of flowing wells was decreasing in the 1920s, led to the theoretical finding of aquifer

compressibility (Meinzer, 1928;Meinzer and Hard, 1925).

Groundwater development in the Dakota aquifer started with flowing wells in the 1880s. After

active drilling in the 1900s, investigations in the 1910s showed that many flowing wells had stopped

flowing. To determine the groundwater budget, a study area of 18 townships near Ellendale, North

Dakota (Ranges 48 through 65 West along Township 129 North) with a total of 320 flowing wells

supplied by the Dakota sandstone was selected. The rate of discharge through flowing wells was

estimated to be close to 3000 gallons per minute during the 38-year period from 1886 to 1923, but the

rate of lateral recharge through eastward percolation was inferred to be less than 1000 gallons per minute

(Meinzer and Hard, 1925). Although these estimates could be very inaccurate, they were sufficient to

demonstrate the excess of discharge through flowing wells over recharge. Meinzer and Hard (1925)

concluded that most of the water discharged through the flowing wells was taken out of storage in the

sandstone aquifer, indicating that the sandstone aquifer was compressible. It was also observed that the

artesian head would increase gradually for some time after a flowing well was shut off, which is a

manifestation of elasticity of the aquifer medium (Meinzer and Hard, 1925).

By summarizing these observations of flowing wells, as well as the evidences of compressibility

and elasticity of compacted sand, land subsidence in an oil field, water level fluctuations produced by

ocean tides, and water level fluctuations produced by railroad trains, Meinzer (1928) concluded that

confined aquifers are compressible and elastic. Although geochemical and numerical studies several

decades later showed that leakage also contributed to well discharge in the Dakota aquifer (Bredehoeft

et al., 1983;Leonard et al., 1983;Swenson, 1968), this did not undermine the role of flowing wells that

had intrigued the interest of hydrogeologists.

Several years later, by assuming that discharge of groundwater from storage as head falls is similar

to release of heat as temperature decreases, Theis (1935) recognized that confined aquifers possess a

property analogous to heat capacity and derived the equation characterizing the transient behavior of

hydraulic head due to discharge of a well. Theis's solution was not understood by the groundwater

hydrology community until Jacob (1940) defined the coefficient of storage as a combination of vertical

compressibility of the porous medium and compressibility of water. Hereafter, numerous efforts were





devoted to determining the aquifer parameters using transient well hydraulics and identifying the

behavior of drawdown/flow rate in other aquifers (leaky aquifers, unconfined aquifers).

**6.2 Transient well hydraulics in flowing wells and non-flowing wells in confined aquifers**

In the early 1930s, the high demand for groundwater led to evaluation of groundwater in different

parts of the United States and pumping tests using the Thiem equilibrium method were conducted to

obtain hydraulic conductivity in several regions (Lohman, 1936;Theis, 1932;Wenzel, 1936).

Unfortunately, it was found difficult to consistently obtain aquifer parameters because of the increasing

drawdown with time (Wenzel, 1936).

To interpret the time-varying drawdown, Charles Vernon Theis assumed that groundwater flow

disturbed by a sink withdrawing water was analogous to heat conduction disturbed by a sink withdrawing

heat and resorted to Clarence Isador Lubin, a mathematician at the University of Cincinnati, for the

solution of temperature distribution of a uniform plate under two different conditions (White and Clebsch,

1993). The first condition is the introduction of a sink kept at 0 temperature, which corresponds to the

constant-drawdown aquifer test problem and is applicable to flowing wells, and the second condition is

the introduction of a sink with a uniform heat flow rate, which corresponds to the constant-rate pumping

test problem. It was fortunate that the solution of the second problem was readily available in the field

of heat conduction (Carslaw, 1921). In this way, Theis (1935) obtained the analytical solution of time-

dependent drawdown induced by pumping and opened the door of determining aquifer parameters using

transient well hydraulics.

When a flowing well has been shut off for a duration of time, upon reopening, the discharge rate

decrease with time, which can be considered as a constant-drawdown aquifer test. Based on Smith (1937)

solution to the analogous problem in heat conduction (the first problem raised by Theis), Jacob and

Lohman (1952) derived a solution to the constant-drawdown well test problem in a confined aquifer

and verified the results based on flowing wells in the Grand Junction artesian basin, Colorado. Several

years later, after the classical work on constant-rate pumping problem in leaky aquifers (Hantush and

Jacob, 1955), Hantush (1959) derived a solution to the constant-drawdown well hydraulics to a flowing

well in leaky aquifers. In fact, constant-drawdown tests can also be carried out in non-flowing wells

either by using a specially designed pump or by connecting the well to a pressurized water container at

the surface (Mishra and Guyonnet, 1992). Such constant-drawdown tests have been found to be





particularly useful in low-permeability aquifers (Jones, 1993;Tavenas et al., 1990;Wilkinson, 1968).

In summary, although the door of transient well hydraulics was directly opened by Theis (1935) based on constant-rate pumping tests, constant-drawdown well tests triggered by flowing wells belong to an indispensable component of transient well hydraulics and are still receiving active attention in the current century (Chang and Chen, 2002;Wen et al., 2011;Tsai and Yeh, 2012;Feng and Zhan, 2019). It is worth noting that current models on transient well hydraulics did not fully account for the relationship between groundwater recharge from precipitation and groundwater discharge in wells, for example, the

higher flow rate in November than that in September shown in Fig. 5a can not be explained by current theories.

## 7 Topographically-controlled flowing wells and topographically-driven flow systems (1940s-current)

### 7.1 Topographically-controlled flowing wells

Based on the principle of the conservation of mass and the laws of thermodynamics, Hubbert (1940) proposed the fundamental rules to obtain graphical solutions of regional groundwater flow. In a homogeneous and isotropic aquifer with a symmetrical topography between two streams, a cross-sectional flow net was drawn by assuming that groundwater recharge is distributed over the whole air-water interface except for the streams at the valleys. Fig. 7 shows that flow lines diverge from recharge

areas and converge toward discharge area at the bottoms of valleys. Fetter (1994) superposed some piezometers onto the equipotential lines, which clearly shows that hydraulic head decreases with well depth in topographic highs near the divide, and increases with well depth in topographic lows near the valley. Moreover, below the streams, hydraulic head in the aquifer is higher than the elevation of ground surface, which constitutes a necessary and sufficient condition of vertical flowing wells. Such flowing

wells in homogeneous unconfined aquifers were termed topographically-controlled flowing wells (Freeze and Cherry, 1979).





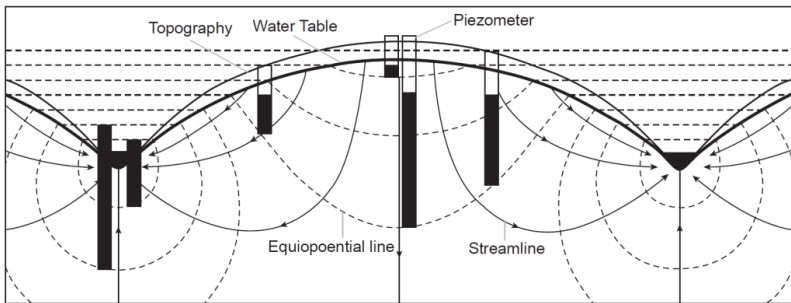

Fig. 7 The flow net of groundwater flow between two river and the head in selected piezometers (Modified from Fetter, 1994).


Although the concept of topographically-controlled flowing wells has been included in some textbooks (Domenico and Schwartz, 1998;Freeze and Cherry, 1979;Heath, 1983;Kasenow, 2010;Lohman, 1972a), the number of which is limited compared with the large number of existing groundwater textbooks. To quantify the occurrence of flowing wells in topographic lows of unconfined aquifers, Wang

et al. (2015b) examined the zone with flowing wells under different water table undulations (characterized by different $\alpha$, which is the ratio of water table undulations to topography undulations) and basin width/depth ratios ($L/|D|$, where $L$ is the basin length and $|D|$ is the basin depth). By fixing $|D|$, increases in $\alpha$ and decreases in $L$ both lead to increased hydraulic gradient between recharge and discharge areas. Based on the distribution of head exceeding surface (termed artesian head in their paper),

it was found that the zone with flowing wells is always within the discharge area and the ratio of its size to the whole basin is proportional to the hydraulic gradient (Fig. 8). Therefore, in homogeneous basins, the hydraulic gradient is the main control factor of flowing wells.

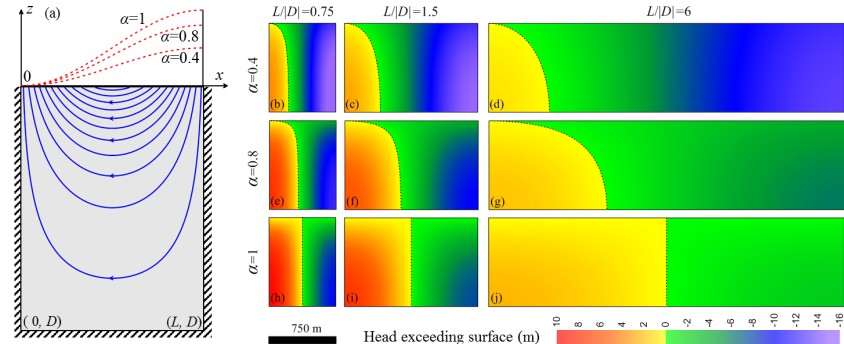

Fig. 8 The geometry of a unit basin with a length of L and depth of |D| (a) and the distribution of head

exceeding surface in the unit basin under different water table undulations (corresponding to different $\alpha$)





and basin length/width ratios (b-j). (Modified from Wang et al., 2015)

**7.2 Topographically-driven groundwater flow systems**

In the 1950s and 1960s, the high demand for water on the Canadian Prairies led to institutional programs of ground water exploration and research. The phenomena anticipated by Hubbert (1940), like

the mean water table closely follows the topography, hydraulic head decreases with well depth in topographic highs (corresponding to recharge areas) and increases with well depth in topographic lows (corresponding to discharge areas), and flowing wells occur in topographic lows, were quite common in the Canadian Prairies (Meyboom, 1962, 1966;Tóth, 1962, 1966). Based on the field observations, two similar but slightly different conceptual models of topographically-induced groundwater flow were

developed by Tóth (1962) and Meyboom (1962), both of which believed that flowing wells could occur in the discharge area (Fig. 9).

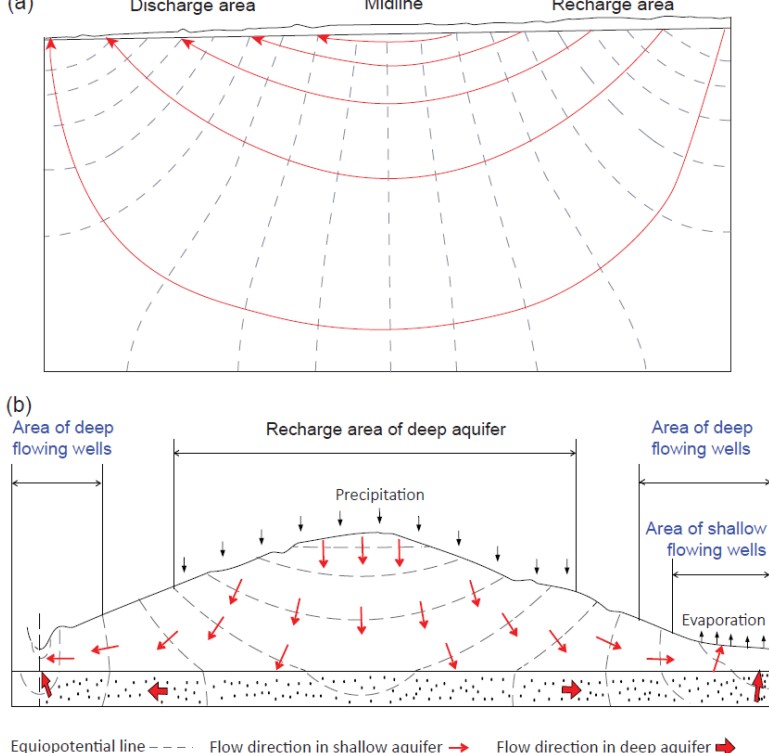

Fig. 9 Topographically-induced flow systems: (a) homogeneous basin (Modified from Toth, 1962) and

(b) heterogeneous basin with a higher permeability layer in the bottom (Modified from Meyboom, 1962).



Because the glacial deposits with low permeability are still efficient for infiltration of rainfall and

evaporation of soil water, Tóth (1962) neglected the permeability difference between glacial materials

and the underlying sandstone. By solving the Laplace equation for a homogeneous unit basin with a

known water table change linearly from the divide to the valley, Tóth (1962) obtained a cross-sectional

flow net (Fig. 9a) slightly different from Hubbert's. It was found that groundwater discharge could cover

the entire lower half of the unit basin, and the whole discharge area has higher hydraulic head than the

corresponding elevation of water table (the cases shown in Fig. 8h,i,j), which fulfills Meinzer's (1923)

definition of "artesian water". In a composite basin whose water table configuration is the superposition

of a sinusoidal curve and a linear regional slope, Tóth (1963) obtained a flow net showing the

simultaneous occurrence of local flow systems and intermediate or regional flow systems across divides

within a large basin. Note that Toth (1963) cited the pattern of cross-sectional groundwater flow inferred

from hydrochemical facies in the northern part of the Atlantic Coastal Plain (Back, 1960;Back, 1966) to

support his finding. Because Back (1960) and Toth (1963) are not directly related to flowing wells, plots

of nested flow systems are not shown here.

        Meyboom (1962) qualitatively plotted the flow net called the "Prairie Profile" (Fig. 9b), which

considered the higher permeability of the sandstone than glacial deposits. The profile has large zones of

shallow or deep flowing wells as well as large zones of evapotranspiration in the discharge area. Although

Toth and Meyboom had disagreement on some specific details, they agreed that the combination of the

two models "*gives a good description of the unconfined region of groundwater flow in the western

Canadian Prairies*"(Tóth, 2005). Several years later, as a PhD student, R. Alan Freeze decided to bring

the ideas of Meyboom and Toth together by numerically simulating steady-state regional groundwater

flow in heterogeneous basins with any desired water table configuration (Freeze, 2012). Freeze and

Witherspoon (1967) demonstrated that heterogeneity does not affect the topographically-induced flow

pattern from recharge to discharge areas (Fig. 10). In fact, Fig. 10 b and 10c indicate that a higher

permeability in the lower aquifer could lead to a larger area of flowing wells. Moreover, they found that

confined aquifers like those shown in Fig. 10b and 10c need not outcrop to produce flowing well

conditions.

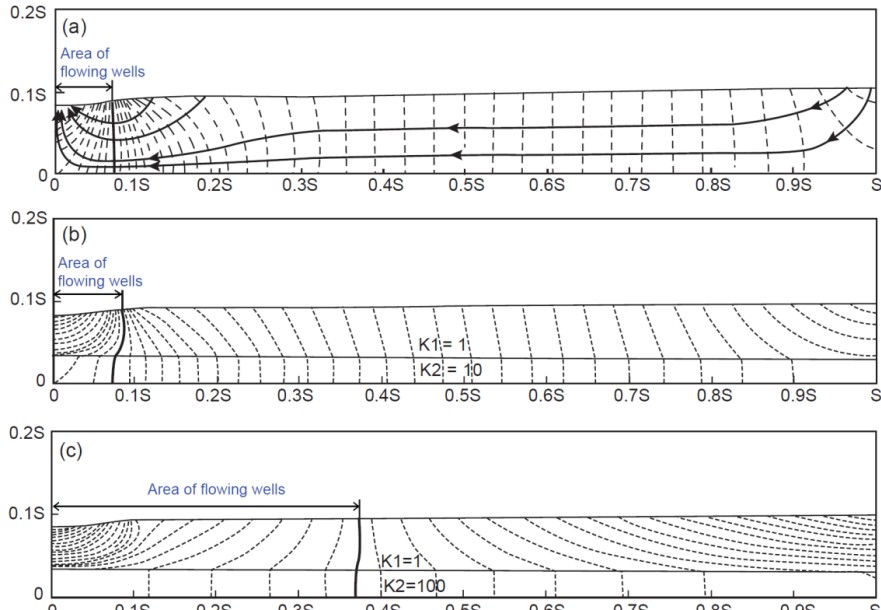

Fig. 10 The distribution of equipotential lines and area of flowing wells in homogeneous (a) and heterogeneous basins (b-c). Also shown in (a) is the streamlines showing the pattern of groundwater flow from the recharge to the discharge area. (Modified from Freeze and Witherspoon, 1967).

**7.3 Hydraulics of topographically-controlled flowing wells**

The Ordos Plateau in northwestern China has a thick Cretaceous sandstone, the thickness of which could be up to around 1000 m. Because the thin Quartnary deposits overlying the Cretaceous sandstone have much higher permeability, and there is no continuous aquitard within the Cretaceous sandstone, the Ordos Plateau can be considered as a thick unconfined aquifer. Due to the undulating topography, there are numerous topographically-controlled flowing wells drilled into the Cretaceous sandstone. For example, in the Wudu lake catchment with an area of around 200 km$^2$, there were 15 flowing wells in 2015 (Wang et al., 2015b). In recent years, more flowing wells have been drilled. The majority of flowing wells in this catchment range in depth from 70 m to 300 m, and one well reaching the bottom of the Cretaceous sandstone has a depth of 800 m. The wells have long-screens and are cased only in the very shallow part corresponding to the Quartnary deposits. It is interesting that groundwater collected at the flowing wells has a hydrochemical facies of Na-HCO$_3$, does not contain NO$_3^-$, and is depleted in $\delta^2$H and $\delta^{18}$O, all of which are quite different from groundwater in recharge areas (Wang et al., 2015a). Moreover, Mg in groundwater collected from flowing wells has been greatly removed by the process of clay





formation, leading to much lower $\delta^{26}Mg$ than samples in recharge areas (Zhang et al., 2018a). These hydrochemical and isotopic evidences show that groundwater collected at the outlets of flowing well could represent deep groundwater and is seldom mixed with shallow groundwater (Zhang et al., 2019).

To examine the hydraulics of topographically-controlled flowing wells, the water exchange between the aquifer and the flowing well has been simulated using the revised multi-node well (MNW2)

package (Konikow et al., 2009) of MODFLOW by considering a flowing well in a three-dimensional unit basin (Zhang et al., 2018b). The hydraulic head of the flowing well, $H_w$, is set equal to the elevation of the ground surface. The trend of increasing hydraulic head with depth in the discharge area results in hydraulic head in the aquifer smaller than $H_w$ in the shallow part and larger than $H_w$ in the deep part. Therefore, there is groundwater inflow from the aquifer to the flowing well ($Q_{in}$) in the deep part, and

groundwater outflow from the flowing well to the aquifer ($Q_{out}$) in the shallow part. If $Q_{in}=Q_{out}$, then flow rate at the well outlet equals 0 (Fig. 11a), which is the same as a non-flowing well in the discharge area as reported in Zinn and Konikow (2007). However, if $Q_{in}>Q_{out}$, flow rate at the well outlet is above 0 (Fig. 11b), which results in water overflow at the surface. In some extreme cases, for example, if the water table coincide with the ground surface, or the shallow part is cased, $Q_{out}$ equals 0 and flow rate at

the well outlet is determined by $Q_{in}$ (Fig. 11c). It has also been found that the simultaneous occurrence of inflow and outflow could occur in a thick confined aquifer (Zhang et al., 2018b). Therefore, the 3rd condition proposed by both de Thury (1830) and Bond (1865), and the 7th condition given by Chamberlin (1885), is not a necessary condition for flowing wells.

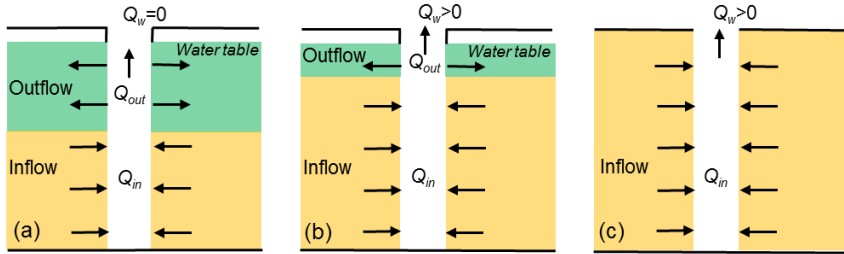


Fig. 11 Conceptual cross-sectional views of flow around non-pumping wells in the discharge area of a basin. (a) A non-flowing well with $Q_{in} = Q_{out}$; (b) a flowing well with $Q_{in} > Q_{out}$; (c) a flowing well with $Q_{in} > Q_{out} = 0$. (Modified from Zhang et al., 2018b)

Based on the plots shown in Fig. 7 through 11, many qualifying conditions of flowing wells proposed in the 19th century have been found to be not necessary conditions since 1940. As shown in Fig.



7 through 10, the pattern of groundwater flow in thick unconfined aquifers is quite different from that in thin unconfined aquifers as well as that in confined aquifers. The concepts of topographically-controlled flowing wells and topographically-induced groundwater flow systems have led to a paradigm shift of hydrogeology (Bredehoeft, 2018;Tóth, 2005). The spatial distribution of groundwater age in thick unconfined aquifers (Jiang et al., 2010;Jiang et al., 2012) is also more complicated than that in a confined aquifer. Although the transient behavior of groundwater flow to geologically-controlled flowing wells has been studied in the 1950s (Hantush, 1959;Jacob and Lohman, 1952), there is no research on the transient groundwater flow to topographically-controlled flowing wells. Moreover, research coupling groundwater recharge from precipitation and groundwater discharge through flowing wells, which is critical to interpret the increased flow rate from September to November as shown in Fig. 5a, is also missing.

## 8 Conclusions and suggestions

The first recorded recognition of flowing wells was as early as 1126 in northern France, but it was the advent of modern cable-tool drilling equipment in Europe in the early 19[th] century that made flowing wells common. In the textbook by Davis and De Weist (1966), it was pointed out that exploration of flowing wells in Europe in the 18[th] century was responsible for stimulating the advancement of water well drilling technology. In fact, flowing wells, which represent a spectacular natural phenomenon of deep groundwater, also instigated the science of groundwater. The pursuit of answers to fundamental questions generated by flowing wells in confined aquifers moved the science forward for more than a century since the early 19[th] century. Moreover, it is interesting that since the 1940s, flowing wells in unconfined aquifers played a significant role on the new paradigm, i.e., a paradigm shift from piston flow in confined aquifers to topographically-driven flow in either homogeneous or heterogeneous basins.

The spectacular flowing wells in Paris and London in the early 19[th] century drew widespread attention to this most noticeable feature of groundwater, which was the early impetus behind the start of groundwater science in the mid 19[th] century. It was not a coincidence that Darcy (1856) did his monumental laboratory experiments soon after he did pipe flow experiments prompted by flowing wells. He was followed by his colleague Dupuit (1863) to develop the hydraulics of steady flow to wells. The term "flowing well" was introduced by Chamberlain (1885) in his classic USGS report, which provided the first comprehensive explanation of flowing wells using hydrogeological principles. Based on field


investigations of the Cambrian-Ordovician aquifer system in Wisconsin, he recognized the role of confining beds in creating flowing well conditions and also that these confining beds are leaky. This was followed soon after by the classic work by Darton (1897, 1905) who studied the regional Dakota aquifer. Meinzer and Hard (1925) and Meinzer (1928) deduced from declining discharge of flowing wells and

excess of discharge over recharge that confined aquifers are elastic and have storage capability related to compressibility. This prompted Theis (1935) of the USGS to initiate unsteady state well hydraulics for non-leaky aquifers, although leakage recognized decades earlier set the stage for Hantush to pioneer the hydraulics of pumping wells and flowing wells in aquifers with leaky confining beds in the 1950s (Hantush and Jacob, 1955; Hantush, 1959).

The wide occurrence of regional scale confined aquifers showing ubiquitous flowing wells in sedimentary rocks in France, England and the United States resulted in confined-aquifer piston flow being a broadly useful conceptualization. However, this resulted in the common misconception that flowing wells must be geologically-controlled, and the confusion of the term "artesian". In his monumental treatise on the theory of groundwater motion, Hubbert (1940) realized flowing wells can

occur in entirely unconfined and homogeneous conditions based only on topographic control. Hubbert's concepts of topographically-driven groundwater flow and topographically-controlled flowing wells were further developed by Toth (1962, 1963, 1966) and Meyboom (1962, 1966) in the Canadian Prairies. Subsequent studies by Freeze and Witherspoon (1967) and Zhang et al. (2018b) found that several qualifying conditions of flowing wells proposed by Chamberlain (1885) are not necessary at all.

Although the theory of topographically-driven groundwater flow systems has been considered to be a paradigm shift of modern hydrogeology (Bredehoeft, 2018;Madl-Szonyi, 2008;Tóth, 2005), the misconception that flowing wells must be geologically-controlled is still impeding the acceptance of the new paradigm. Moreover, many modern textbooks still have not fully clarified the differences and implications of the two types of flowing wells, geologically versus topographically controlled. Therefore,

consistent terminology and a complete description of both types of flowing wells are expected in future groundwater textbooks.

Based on the summary of the role of flowing wells on the evolution of many concepts and principles of groundwater hydrology, it is desirable that integrating the root of flowing wells into textbooks and courses of groundwater hydrology would inspire the interest of beginners, and also lead to a deeper

understanding of the science of groundwater (Deming, 2016). Although the number of flowing wells has





decreased significantly since the 20[th] century, flowing wells still occur widely in many parts of the world and the overflow of some flowing wells have lasted for more than a century. The aquifer hydraulics behind the sustainability of flowing wells, i.e., coupling groundwater discharge in flowing wells to groundwater recharge from precipitation, deserve further research. Moreover, because geologically-

controlled flowing wells usually occur in topographical lows, a generalized theory on the simultaneous control of topography and confining bed on the occurrence of flowing wells is expected in the future.



**Code/Data availability**

Not apply to the review paper.


**Author contribution**

XJ and JC prepared the manuscript with contributions from LW.

**Competing interests**

The authors declare that they have no conflict of interest.

**Acknowledgements**

This study was supported by the Teaching Reform Funds of China University of Geosciences and

the National Program for Support of Top-notch Young Professionals. The authors thank Vitaly Zlotnik

of University of Nebraska-Lincoln for discussion on the ambiguous term "artesian" and Eileen Poeter of

Colorado School of Mines for suggestions on improving the manuscript.



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
