# Peer review of "Flowing wells: history and role as a root of groundwater hydrology"

_Hydrology and Earth System Sciences, 2020_

## Referee Comment (RC1) · Anonymous Referee #1 · 5 Aug 2020

July 25, 2020

Review of "Flowing wells: history and role as a root of groundwater hydrology by Xiao-Wei Jiang, John Cherry, and Li Wan, MS No.: hess-2020-270 for Special Issue: History of hydrology (HESS/HGSS inter-journal SI)

1. Granted, this is supposed to be a "review article," but I found it to be hopelessly long, rambling, and repetitive. The text is more suited to a lengthy technical report or a first draft of a book.

2. I find that I am in disagreement with the main thesis of the article. That is, that the study of "flowing wells since the early 19th century led to the birth of many fundamental concepts and principles of groundwater hydrology." The reason is that the physical laws

and geology that control groundwater flow are the same for flowing and non-flowing wells.

3. There are several pages devoted to a discussion of the definition of the term "artesian." While it is true that the literature on this is conflicting and worthy of review and discussion, it's an example of how this manuscript rambles everywhere in its discussion. It lacks focus.

4. The authors claim that it is a "misconception that flowing wells must be geologically-controlled" (line 749). Seems to me that all groundwater flow is affected by the geology and is therefore "geologically controlled." Perhaps what is needed here is a straightforward definition of what the authors mean by the term "geologically controlled."

Summary Recommendation: Accept with major revision. If the editors want to publish a lengthy paper, that decision is up to them. In any case, my opinion is that the paper needs focus and condensation.

---

## Referee Comment (RC2) · Garth van der Kamp (Referee) · 6 Aug 2020

This manuscript provides a detailed and global overview of the progress of understanding of flowing wells. In this sense it has the potential to make a worthwhile contribution to the science of hydrogeology and its historical development. Flowing wells have always been, and still are, a topic of wonderment and questioning.

In such a historical overview, spanning many countries and two centuries, it is a formidable challenge to recognize and give due credit to all significant contributions, the literature of which may not be easily accessible. In this case the authors would do well to widen their search and in particular to refer to the historical overview by J. J. de Vries (2007), which describes several pre-1940 examples of what the present paper describes as "topographically-driven" groundwater flow. (History of Groundwater Hydrology. In: The handbook of Groundwater Engineering (Jacques W. Delleur ed.), Chapter 1: p.1.1-1.39. CRC Press, 2007).

More generally this manuscript is perhaps over emphasizing the analysis of "topographically-driven" groundwater flow in the 1960's by Toth and others as a paradigm shift. Such a shift in science is typically defined as, for example "an important change that happens when the usual way of thinking about or doing something is replaced by a new and different way" [https://www.merriam-webster.com/dictionary/paradigm%20shift]. It might be added that in science such a paradigm shift also opens up new and expanded fields of enquiry. Casting historical developments in hydrogeology in terms of paradigm shifts can be a valuable and enlightening exercise by emphasizing when and how new concepts were developed and became accepted. The authors are encouraged to maintain a paradigm perspective.

With regard to topographically-driven flow the manuscript makes clear that it was recognized very early on, in the 19th century, that the occurrence of flowing wells is dependent on a recharge source of the groundwater at a higher elevation. In other words, topographically-driven groundwater flow (together with various geological configurations) has always been recognized. It was also recognized early on that flowing wells can occur even where there is no overlying low-permeability formation confining an aquifer (See for example the measurement-based flow diagrams published by Pennink in 1907 as reproduced by de Vries (2007, p. 1-11). As pointed out in the manuscript (L 60) Hubbert noted this (Hubbert, 1940, p.913: "As a matter of fact, in order to have artesian potentials, an aquifer need not be overlain by impermeable material, and it need not out-crop.")

Clearly there is a continuum of hydrogeological conditions that can result in flowing wells, varying from highly confined aquifers with distant higher-elevation outcrops to entirely homogeneous surficial aquifers, the latter likely relatively rare. Thus a distinction between "topographically-controlled" and "geologically controlled" flow may not be

useful. Incidentally the Paskapoo formation in southern Alberta which inspired Toth's early analyses is far from constituting " a thick unconfined aquifer (L. 335), but "represents a foreland deposit of a siltstone and mudstone-dominated fluvial system" with interspersed channel deposits of coarse-grained sandstone (Grasby et al., 2008. Can J Earth Sc., 45: 1501-1516, p.1).

Toth and others set off an important change in hydrogeology by introducing rigorous mathematical description of regional groundwater flow, at first by calculus methods, soon followed by numerical modelling. Numerical modeling has become a major and standard tool in hydrogeology. Its introduction by Toth, followed up by Freeze, could indeed be characterized as a paradigm shift, in the sense of doing something in a new and different way, which has also led to new research and discovery. But their analysis and description, as such, of flowing wells occurring without an overlying aquitard does not appear to merit characterization as a paradigm shift because this idea was already present in the literature and was recognized as a logical consequence of Darcy's Law.

Perhaps a case could be made that the recognition of aquifer compressibility by Meinzer and others, based on analysis of flowing well yields, was the beginning of a paradigm shift in hydrogeology. The manuscript describes how the follow-up work by Theis (1935) and Jacob (1940) led to a widening field of enquiry and understanding of transient groundwater hydraulics, particularly with regard to well hydraulics.

---

## Author Comment (AC1) · 13 Aug 2020

Review of "Flowing wells: history and role as a root of groundwater hydrology by Xiao-Wei Jiang, John Cherry, and Li Wan, MS No.: hess-2020-270 for Special Issue: History of hydrology (HESS/HGSS inter-journal SI)

1. Granted, this is supposed to be a "review article," but I found it to be hopelessly long, rambling, and repetitive. The text is more suited to a lengthy technical report or a first draft of a book.

Response: Thanks for the suggestion. We will condense the paper.

2. I find that I am in disagreement with the main thesis of the article. That is, that the study of "flowing wells since the early 19th century led to the birth of many fundamental

concepts and principles of groundwater hydrology." The reason is that the physical laws and geology that control groundwater flow are the same for flowing and non-flowing wells.

Response: We agree that many physical laws that control groundwater flow are the same for flowing and non-flowing wells. However, flowing wells, which have visible groundwater at the surface, always attracts the interest of people. As we demonstrated in the manuscript, the concepts stemmed from flowing wells in confined aquifers include permeability and compressibility, while the principles include Darcy's law, role of aquitards on flowing well conditions and the piston flow pattern, steady-state well hydraulics in confined aquifers, and transient well hydraulics towards constant-head wells in confined or leaky aquifers. Because all of these principles are applicable even if flowing well conditions have disappeared, how these principles were initially developed had been forgotten by many groundwater hydrologists. As stated in the advertisement of the current special issue History of Hydrology, "As a hydrological community, we are keen to further our science, and it is therefore of utmost importance to understand what the roots of our science are." It is desirable that integrating the root of flowing wells into textbooks and courses of groundwater hydrology would inspire the interest of beginners, and also lead to a deeper understanding of the science of groundwater (Deming, 2016).

3. There are several pages devoted to a discussion of the definition of the term "artesian." While it is true that the literature on this is conflicting and worthy of review and discussion, it's an example of how this manuscript rambles everywhere in its discussion. It lacks focus.

Response: As pointed out by the reviewer, it is worth reviewing and discussing the conflicting usage in the literature. In our opinion, the conflicting usage of "artesian" is one reason that leads to neglecting the role of flowing wells on the evolution of groundwater hydrology. Therefore, we use several pages to discuss the birth and confusing usage of the term "artesian". To make the paper focus, we will decide whether to remove this

part or put this part in the Supplement.

4. The authors claim that it is a "misconception that flowing wells must be geologically controlled"(line 749). Seems to me that all groundwater flow is affected by the geology and is therefore "geologically controlled." Perhaps what is needed here is a straightforward definition of what the authors mean by the term "geologically controlled."

Response: We totally agree with the referee that all groundwater flow is affected by the geology and is therefore "geologically controlled." In fact, this term was initially defined by Freeze and Cherry (1979). We have defined this term in Lines 153-155 (To differentiate the two types of flowing wells due to different causes, Freeze and Cherry (1979) defined geologically-controlled flowing wells and topographically-controlled flowing wells. As shown in Fig. 1, the former develop in confined aquifers and receive recharge at upland outcrops, while the latter occur in the topographic lows of unconfined aquifers.")

Summary Recommendation: Accept with major revision. If the editors want to publish a lengthy paper, that decision is up to them. In any case, my opinion is that the paper needs focus and condensation.

Response: Thanks very much! We will focus and condense the paper.

References:

Freeze, R. A., and Cherry, J. A.: Groundwater, Prentice-Hall. Inc., Englewood Cliffs, N.J., 1979.

Deming, D.: The Importance of History, Groundwater, 54, 745-745, 10.1111/gwat.12458, 2016.
* * *

---

## Author Comment (AC2) · 13 Aug 2020

Response to Referee #2 (Garth van der Kamp)

This manuscript provides a detailed and global overview of the progress of understanding of flowing wells. In this sense it has the potential to make a worthwhile contribution to the science of hydrogeology and its historical development. Flowing wells have always been, and still are, a topic of wonderment and questioning.

1. In such a historical overview, spanning many countries and two centuries, it is a formidable challenge to recognize and give due credit to all significant contributions, the literature of which may not be easily accessible. In this case the authors would do well to widen their search and in particular to refer to the historical overview by

J. J. de Vries (2007), which describes several pre-1940 examples of what the present paper describes as "topographically-driven" groundwater flow. (History of Groundwater Hydrology. In: The handbook of Groundwater Engineering (Jacques W. Delleur ed.), Chapter 1: p.1.1-1.39. CRC Press, 2007).

Response: Thanks very much for suggesting this reference. We benefit a lot from this reference. We also found more references from the review by de Vries.

2. More generally this manuscript is perhaps over emphasizing the analysis of "topographically-driven" groundwater flow in the 1960's by Toth and others as a paradigm shift. Such a shift in science is typically defined as, for example "an important change that happens when the usual way of thinking about or doing something is replaced by a new and different way" [https://www.merriamwebster.com/dictionary/paradigm%20shift]. It might be added that in science such a paradigm shift also opens up new and expanded fields of enquiry. Casting historical developments in hydrogeology in terms of paradigm shifts can be a valuable and enlightening exercise by emphasizing when and how new concepts were developed and became accepted. The authors are encouraged to maintain a paradigm perspective.

Response: Thanks for the encouragement of maintaining a paradigm perspective. We will incorporate the referee's suggestions on the earlier references to alleviate the problem of "over emphasizing" the contributions by Toth and others.

3. With regard to topographically-driven flow the manuscript makes clear that it was recognized very early on, in the 19th century, that the occurrence of flowing wells is dependent on a recharge source of the groundwater at a higher elevation. In other words, topographically-driven groundwater flow (together with various geological configurations) has always been recognized. It was also recognized early on that flowing wells can occur even where there is no overlying low-permeability formation confining an aquifer (See for example the measurement-based flow diagrams published by Pennink

in 1907 as reproduced by de Vries (2007, p. 1-11). As pointed out in the manuscript (L 60) Hubbert noted this (Hubbert, 1940, p.913: "As a matter of fact, in order to have artesian potentials, an aquifer need not be overlain by impermeable material, and it need not out-crop.")

Response: Thanks very much for the suggestions! We totally agree with the referee's statement that "the occurrence of flowing wells is dependent on a recharge source of the groundwater at a higher elevation", which had been realized by Bernardino Ramazzini (1691) and other European researchers in the 1800s, can be considered as an early recognition of topographically-driven groundwater flow. After reading J. J. de Vries (2007a) and J. J. de Vries (2007b), we realize that King (1899) and Pennink (1905) had already described the phenomenon of upward flow in discharge areas, which is critical for flowing wells. This understanding of artesian water was 35 years earlier than Hubbert (1940). We will incorporate them in the revision.

4. Clearly there is a continuum of hydrogeological conditions that can result in flowing wells, varying from highly confined aquifers with distant higher-elevation outcrops to entirely homogeneous surficial aquifers, the latter likely relatively rare. Thus a distinction between "topographically-controlled" and "geologically controlled" flow may not be useful. Incidentally the Paskapoo formation in southern Alberta which inspired Toth's early analyses is far from constituting "a thick unconfined aquifer (L. 335), but "represents a foreland deposit of a siltstone and mudstone-dominated fluvial system" with interspersed channel deposits of coarse-grained sandstone (Grasby et al., 2008. Can J Earth Sc., 45: 1501-1516, p.1).

Response: We agree with the referee that "there is a continuum of hydrogeological conditions that can result in flowing wells, varying from highly confined aquifers with distant higher-elevation outcrops to entirely homogeneous surficial aquifers". A similar statement "The variety of hydrologic situations that will give rise to potentials below the ground surface larger than those a few feet above is very great" was given by Hubbert (1940). Hubbert (1940) also pointed out that the extensive textbook illustration of flowing wells in confined aquifers has "obscured a more complete understanding of arte-
sian phenomena". Therefore, in the classical textbook written by Freeze and Cherry
(1979), flowing wells were divided into "geologically-controlled" and "topographically-
controlled".

Freeze and Cherry (1979) pointed out that "The primary control on flowing wells is
topography", implying that geologically-controlled flowing wells, i.e., flowing wells in
confined aquifer, are also controlled by topography. Therefore, we agree that a dis-
tinction between topographically-controlled and geologically controlled flowing wells is
not enough. In fact, we have pointed out in the final sentence of the conclusion that
"because geologically-controlled flowing wells usually occur in topographical lows, a
generalized theory on the simultaneous control of topography and confining bed on the
occurrence and the hydraulics of flowing wells is expected in the future." We hope the
current paper would inspire studies in this field by combining Toth's "rigorous mathe-
matical description of regional groundwater flow" and classical well hydraulics which
assumes a totally flat initial hydraulic head field.

5. Toth and others set off an important change in hydrogeology by introducing rigorous
mathematical description of regional groundwater flow, at first by calculus methods,
soon followed by numerical modelling. Numerical modeling has become a major and
standard tool in hydrogeology. Its introduction by Toth, followed up by Freeze, could
indeed be characterized as a paradigm shift, in the sense of doing something in a new
and different way, which has also led to new research and discovery. But their analysis
and description, as such, of flowing wells occurring without an overlying aquitard does
not appear to merit characterization as a paradigm shift because this idea was already
present in the literature and was recognized as a logical consequence of Darcy's Law.
Response: We agree with the reviewer that Toth and others' analysis and description
of flowing wells occurring without an overlying aquitard, which had already been de-
scribed by Pennink (1905) and Hubbert (1940), does not merit characterization as a
paradigm shift. We will avoid the misunderstanding that "Topographically-controlled

flowing well analyzed by Toth and others is a paradigm shift" in the revision. We also totally agree with the referee that the introduction of rigorous mathematical description of regional groundwater flow by Toth and Freeze could be characterized as a paradigm shift. Anderson (2008) made a similar statement that "the Toth model is an important early exploration of the analysis of regional flow systems". Bredehoeft (2018) pointed out that "Toth's conceptual model of groundwater flow" constitutes a paradigm shift. In the current paper, what we want to emphasize is "topographically-driven groundwater flow systems" is a paradigm shift, which is based on Toth's (2005) statement that "The (topographically-driven) groundwater flow system has become a generally accepted paradigm". In our opinion, Toth's term of "topographically-driven groundwater flow systems" is similar to the referee's term of "rigorous mathematical description of regional groundwater flow", Anderson's (2008) term of "the Toth model", and Bredehoeft's term of "Toth's conceptual model of groundwater flow". We will make it clear in the revision. Like all paradigm shifts, the beginning of the paradigm shift began much earlier, based on field evidences, i.e., King (1899) and Pennink (1905) described the phenomenon of increasing head with depth in the discharge area due to topographically-driven groundwater flow, and the mathematical foundation given by Hubert (1940). However, like nearly all important paradigm shifts in the geosciences, more field evidences had to be accumulated until the new paradigm was overwhelming. The first textbook to systematically introduce Toth's mathematical model of regional groundwater flow systems and differentiate "topographically-controlled" between "geologically-controlled" groundwater flow was the one by Freeze and Cherry (1979), which was published 80 years later than King's (1899) first field description.

6. Perhaps a case could be made that the recognition of aquifer compressibility by Meinzer and others, based on analysis of flowing well yields, was the beginning of a paradigm shift in hydrogeology. The manuscript describes how the follow-up work by Theis (1935) and Jacob (1940) led to a widening field of enquiry and understanding of transient groundwater hydraulics, particularly with regard to well hydraulics.

[Figure]

Response: We agree that the recognition of aquifer compressibility by Meinzer and others, based on analysis of flowing well yields, was the beginning of a paradigm shift in hydrogeology. This paradigm shift had been accepted by the groundwater hydrology community since the 1940s. It is interesting that it took much shorter times for this paradigm shift to become well established in groundwater hydrology.

References:

Anderson, M. P.: Groundwater, Benchmark Papers in Hydrology, IAHS Press, Oxford-shire, 2008.

Bredehoeft, J. D.: The Toth Revolution, Groundwater, 56, 157-159, 10.1111/gwat.12592, 2018.

de Vries J. J.: History of Groundwater Hydrology. In: The handbook of Groundwater Engineering. Edited by J. W. Delleur. CRC Press, 1.1-1.39, 2007a.

de Vries J. J.: Groundwater. In: Geology of the Netherlands. Edited by Th. E. Wong, D.A.J. Batjes & J. de Jager. Royal Netherlands Academy of Arts and Sciences, 295–315, 2007b.

Freeze, R. A., and Cherry, J. A.: Groundwater, Prentice-Hall. Inc., Englewood Cliffs, N.J., 1979.

Hubbert, M. K.: The Theory of Ground-Water Motion, The Journal of Geology, 48, 785-944, 10.1086/624930, 1940.

Jacob, C. E.: On the flow of water in an elastic artesian aquifer, Eos, Transactions American Geophysical Union, 21, 574-586, 10.1029/TR021i002p00574, 1940.

King, F. H.: Principles and conditions of the movements of groundwaters. In 19th Annual Report of the U.S. Geological Survey, part 2, 59–295, 1899.

Pennink, J. M. K.: Over de beweging van grondwater. De Ingenieur 20, 482–492, 1905.

Ramazzini B.: de Fontium mutinensium amiranda scaturigine tractatus physico-hydrostaticus. Cramer and Perachon, Genevae, 1691.

Theis, C. V.: The relation between the lowering of the Piezometric surface and the rate and duration of discharge of a well using ground-water storage, Eos, Transactions American Geophysical Union, 16, 519-524, 10.1029/TR016i002p00519, 1935.

Tóth, J.: The Canadian School of Hydrogeology: History and Legacy, Groundwater, 43, 640-644, 10.1111/j.1745-6584.2005.0086.x, 2005.
* * *
HESSD

---

## Author Response (AR1)

**Response to the Editor**

hess-2020-270, Flowing wells: history and role as a root of groundwater hydrology

Authors: Xiao-Wei Jiang, John Cherry, and Li Wan

Review conclusion, Special issue handling editor: Okke Batelaan

Dear Authors,

As handling-editor for this special issue I like to thank you for your contribution. We now have received two reviews for your manuscript (MS), and you have replied on their comments. I appreciate the thoughtful comments of the reviewers and your replies. The MS is certainly of interest and value for this special issue.

Having gone over the MS myself, and the raised comments, I do agree with these comments and believe that the paper would become clearer, easier to read and more impactful if you would consider the raised comments in a revised MS.

Especially, I would like to stress the following points:

1. The paper can indeed be shortened and written more concise (referee 1). In addition, I notice some unnecessary repetition in the text. This all requires a very careful re-evaluation of each paragraph and what it adds to the total story (and if not already mentioned before). Some examples (but this is certainly not all) are:

-abstract, line 16 and further is an example of the style of writing that you should try to avoid. Another example is line 46-54, extremely long, difficult to understand sentence.

-the introduction can certainly be shortened; it goes into too much detail of aspects that are later (again) discussed.

-section 2 is noted (referee 1) to benefit from more focus. I would keep it still in the MS but shorten it, and maybe restructure it.

-section 7 is very long and would benefit from trimming.

Response: The title has been modified and many parts, including the abstract, Sect. 1, Sect. 2 and Sect. 7 have been shortened.

Specifically, the sentence starting from line 16 in the abstract has been deleted. Two paragraphs in the introduction part are deleted, including the paragraph with line 46-54. To make it more focus, Section 2 has been shortened and restructured by deleting a subsection.

Some redundant words in Sect. 7 have been deleted.

2. Referee 1 does not agree with the main thesis of the paper that "flowing wells… led to the birth of many fundamental concepts and principles of groundwater hydrology". Although, I can see the merit and value in raising attention for this thesis, I am afraid that the style of presentation of the text and building up the arguments for this thesis have not helped the MS in providing a convincing thesis. Again, the above-mentioned line 46-54 is an example. Reconsidering how in the introduction to state the thesis and in the remaining text to build up the arguments for this thesis, requires some careful restructuring of the text.

Response: Thanks for the suggestion. To make the manuscript logical and convincing, we have modified the title and the abstract, completely rewrote the introduction, shortened section 2, added more references in sections 3, 4 and 7, and improved the language.

We have completely rewrote the first paragraph of the introduction part, deleted paragraphs 2 and 3 which go into too much detail of sections 4 through 7, and modified paragraphs 4 and 5. After deleting two paragraph in the introduction part, the plot showing four threads of evolution of physical hydrogeology stemmed from flowing wells is placed in Fig. 1.

The main thesis of the current manuscript is flowing wells in confined aquifers led to developments of three threads, which are elaborated in sections 4, 5 and 6, and flowing wells in unconfined aquifers have contribution to development of the fourth thread, which are elaborated in section 7. These four sections correspond to "the role in the evolution of groundwater science" in the title. Before introducing the role, we use section 2 to introduce the terminology, and use section 3 to introduce the history. We believe that the slightly modified title corresponds to the structure of the manuscript.

In the section 3, 4 and 7, we added some references in the 1600s and 1700s (Ramassini, 1691; Valniseri, 1726) on early thinking of causes of flowing wells, and some references (King, 1899; Pennink, 1905) at the turn of $20^{th}$ century on topographically-driven groundwater flow, and the reference of Versluys (1930) which explicitly demonstrated that aquitards are not necessary conditions of flowing wells by calculation based on the analogy between temperature and hydraulic head.

Moreover, we invited Prof. John F Hermance of Brown University to thoroughly edit the

manuscript.

3. Referee 2, point 1 and 3. Indeed this are important references, which should be investigated for their value and inclusion in this MS.

Response: Following the suggestion of referee #2, we read de Vries (2007) and learned more on the history of hydrogeology. Following de Vries (2007), we have incorporated such references as Ramassini (1691) from Italy, King (1899) from US and Pennink (1905) from Netherland.

In the process of revising, we found Versluys (1930) from Netherland explicitly demonstrated that aquitards are not necessary conditions of flowing wells by calculation based on the analogy between temperature and hydraulic head, which was 10 years earlier than Hubbert (1940) to explicitly point out a similar statement.

4. Referee 2, point 2, 5 and 6. I believe that in recent decades we start to use to easily the word 'paradigm' to try to stress the importance of a certain 'progress in science'. As it is the goal of all research to make some level of progress, we need to be careful when to use such a word, i.e. inflation of the meaning might occur quickly. I believe that the referee is trying to tell you that *if* you use 'paradigm', you should clearly articulate the arguments why it is a paradigm shift.

Response: The main thesis of the current manuscript is flowing wells in confined aquifers led to developments of three threads of physical hydrogeology (sections 4, 5 and 6), while flowing wells in unconfined aquifers have contribution to the theory of topographically-driven groundwater flow systems, which is one thread of physical hydrogeology (section 7).

In the revision, we rewrote the following paragraph in subsection 7.2 to emphasize why quantitative analysis of topographically-driven groundwater flow systems is a paradigm shift, and how others evaluate this paradigm shift in different words.

"As illustrated above, although horizontal flow dominates when the basin width/depth ratio is high and/or hydraulic conductivity is the deep layer is much higher, vertical components of groundwater flow are widespread in either thick unconfined aquifers or aquitards overlying confined aquifers, which is quite different from the flow pattern shown in Fig. 2.1 and 4. The spatial distribution of groundwater age in thick unconfined aquifers is also more complicated

than that in a confined aquifer (Jiang et al., 2010;Jiang et al., 2012). Therefore, quantitative analysis of the topographically-driven groundwater flow systems became a paradigm shift of hydrogeology. This paradigm shift has been expressed by others in similar words. Anderson (2008) comments that "The Tóth model is an important early exploration of the analysis of

95    regional flow systems." Bredehoeft (2018) points out that "Tóth's conceptual model of groundwater flow" represents the beginning of a new era in hydrogeology and termed the paradigm shift to be "the Tóth revolution".

5. Referee 1, point 4 and Referee 2, point 4 both raise the issue of geological vs topographical

100   control. My reading of their comments is that a more critical evaluation of the limitations of this differentiation in your MS, would increase the learnings achieved from your manuscript (i.e. learn from the past is a goal of this special issue).

Response: Freeze and Cherry (1979) gave a comprehensive description of flowing wells in both confined and unconfined aquifers and termed the former to be geologically-controlled and the

105   latter to be topographically-controlled. We agree with the reviewers that in geologically-controlled flowing wells, topography is still the driving force of groundwater flow from the topographic highs to topographic lows.

To avoid confusion, in the revision, we utilize the terms proposed by Toth (1966), confined-flow flowing wells and unconfined-flow flowing wells. We also explicitly point out

110   that "confined-flow flowing wells and unconfined-flow flowing wells as two end members of flowing wells" because different forms of heterogeneity is widespread in the field.

Minor correction: it is 'Davis and De Wiest', Wiest with 'ie'.

Response: Thanks for pointing out the typo.

115   I look forward to your revised MS.

**A list of major changes**

1. To make the structure of manuscript clear, the title has been changed to: "Flowing wells: terminology, history and the role in the evolution of groundwater science". Terminology corresponds to section 2, history corresponds to section 3, and the role in the evolution of groundwater science corresponds to sections 4 through 7.

2. The introduction part has been shortened, and the first paragraph has been completely rewritten.

3. Section 2 has been restructured and shortened.

4. In section 3, the histories of flowing wells in two more countries, Italy and China, are added.

5. In section 4, we add the study of Ramazzini (1691) on flowing wells in Modena, Italy.

6. Section 7 has been significantly revised by deleting some redundant sentences, adding some earlier references on topographically-driven flow (King, 1899; Pennink, 1905) and flowing wells in basins without aquitards (Versluys, 1930) , and rewriting the description of the paradigm shift, i.e., quantitative analysis of topographically-driven flow systems.

7. The conclusions part has been rewritten.

8. The language has been thoroughly improved.

**Flowing wells: terminology, history and the role in the evolution of groundwater science**

Xiao-Wei Jiang[1,*], John Cherry[2], Li Wan[1]

1. MOE (Ministry of Education) Key Laboratory of Groundwater Circulation and Evolution, China University of Geosciences, Beijing 100083, China

2. G360 Institute for Groundwater Research, University of Guelph, Guelph, Ontario N1G 2W1, Canada

*Correspondence to: Xiao-Wei Jiang (jxw@cugb.edu.cn)

**Abstract**

The gushing of water from flowing wells attracted public attention and scientific curiosity as early as early as the 17th century  but little attention has been paid to the influence  of flowing wells on the evolution   of groundwater science. This study asserts that questions posed by  flowing wells since the early 19[th] century led to the birth of many fundamental concepts and principles of  physical hydrogeology. ~~The concepts stemmed from flowing wells in confined aquifers include permeability and compressibility, while the principles include Darcy's law, role of aquitards on flowing well conditions and the piston flow pattern, steady-state well hydraulics in confined aquifers, and transient well hydraulics towards constant-head wells in confined or leaky aquifers, all of which are applicable even if flowing well conditions have disappeared.must be geologically controlledThe occurrence of flowing wellsin topographic lows of unconfined aquifers,was anticipatedin1940~~20[th] 
[revised manuscript text omitted]

1470

---

## Author Response (AR3)

Dear Editor,

Thank you for providing detailed guidance for our manuscript hess-2020-270 entitled "*Flowing wells: terminology, history and the role in the evolution of groundwater science*". We have considered all of the suggestions given by the referee. A detailed response to the referee is attached. We also improved the manuscript by making necessary modifications.

We now submit the revised manuscript for possible publication. If there is any problems, please let me know.

Thank you for your time and consideration of our work. I look forward to hearing from you.

Best regards, Xiao-Wei Jiang Professor of Hydrogeology

**Reponses to Referee #2**

Suggestions for revision by Garth van der Kamp

This manuscript provides a thorough and detailed summary of how the phenomenon of flowing wells has influenced the development of hydrogeological thinking over the centuries. As such it makes a solid contribution to understanding of the history of hydrogeology.

A few minor revisions are recommended to enhance clarity of the manuscript.

1. The terms "aquifer", "confined", and "unconfined" are used throughout the manuscript to distinguish two distinct classes of permeable formations. However, these terms are not explicitly defined in the manuscript. Clear unambiguous definitions should be included because these terms are not always used consistently in the hydrogeological literature. There is also a connection with the definition of the term "artesian" which is discussed extensively in this manuscript. In particular consideration should be given to whether the terms "confined" and/or "artesian" include permeable formations which are bounded above and below by aquitards but in which the water table exists below the bottom of the overlying aquitard. The definitions provided by the classic Freeze and Cherry (1979) text are pertinent in this regard and could be adopted, except that then the terms "confined" and "artesian" are synonymous.

Response: Thanks for the suggestion. According to Freezing and Cherry (1979), a confined aquifer is an aquifer that is confined between two aquitards, while an unconfined aquifer is an aquifer in which the water table forms the upper boundary. Because an aquifer is defined as a saturated permeable geologic unit that can transmit significant quantities of water under ordinary hydraulic gradients, when the water table exists below the bottom of the overlying aquitard, the aquifer belongs to an unconfined aquifer. We have added the definitions of "confined aquifer" and "unconfined aquifer" in Lines 47-48: "Following Freeze and Cherry (1979), a confined aquifer is a saturated permeable geologic unit that is confined between two aquitards, while an unconfined aquifer is the saturated part of a permeable geologic unit in which the water table forms the upper boundary."

2. LL 396-424 (The birth of Darcy's Law). In this section it is clear by modern understanding that the head loss in the aquifer was not the "head loss from the recharge area to the orifice" (L 415). It would likely take years or more for a new hydraulic gradient to be established between the recharge area and the newly changed elevation of the orifice. Presumably Darcy's measurements of flow rate for different elevations of the discharge orifice were taken after the flow rates appeared to be reasonably stabilized. The manuscript should make clear what Darcy's assumptions were in this respect and evaluate these in terms of present-day understanding of the theory of constant-head (drawdown) tests for confined aquifers.

Response: We agree that head loss in the aquifer should not be the "head loss from the recharge area to the orifice", but is the head loss from a position in the aquifer with constant head away from the flowing well to the discharge orifice. We have clarified this in Line 416.

We also added a sentence "From the angle of a constant-head well test, the first part of the left-hand side of Eq. (3b) is the difference in aquifer head loss at the well created by changing the elevation of the discharge orifice, and the second part is the difference in the well loss from riser pipes with different flow distances." in Lines 424-427.

3. LL 607-610. The manuscript makes an intriguing observation with reference to the classic and influential paper by Freeze and Witherspoon (1967): "However the computing capabilities at the time limited the permeability contrasts between aquifer and aquitard units to a factor of 100, which is far from what is reality in actual flow systems in the field and these simulations are then to some degree misleading in light of what we now know." Was this limitation in fact acknowledged by Freeze and Witherspoon, or was it recognized and published afterwards (if so provide the reference), or is this an observation by the authors of the present manuscript and if so, on what basis? This is important because the Freeze and Witherspoon paper continues to play an influential role in the teaching of flow systems and in the formulation of conceptual hydrogeological models.

Response: Thanks very much for raising this question. We have checked with Allan

Freeze and found our statement was not correct. In fact, Freeze and Witherspoon (1967) had considered a case of permeability contrasts between aquifer and aquitard to a ratio of 1000. Therefore, we have deleted the sentence "
[revised manuscript text omitted]